



# The Making of the New European Wind Atlas, Part 2: Production and Evaluation

Martin Dörenkämper[1,*], Bjarke T. Olsen[2,*], Björn Witha[3,4], Andrea N. Hahmann[2], Neil N. Davis[2], Jordi Barcons[5], Yasemin Ezber[6], Elena García-Bustamante[7], J. Fidel González-Rouco[8], Jorge Navarro[7], Mariano Sastre-Marugán[8], Tija Sīle[9], Wilke Trei[3], Mark Žagar[10], Jake Badger[2], Julia Gottschall[1], Javier Sanz Rodrigo[11], and Jakob Mann[2]

[1]Fraunhofer Institute for Wind Energy Systems, Oldenburg, Germany
[2]Wind Energy Department, Technical University of Denmark, Roskilde, Denmark
[3]ForWind, Carl von Ossietzky University Oldenburg, Germany
[4]energy & meteo Systems GmbH, Oldenburg, Germany
[5]Barcelona Supercomputer Center, Barcelona, Spain
[6]Eurasia Institute of Earth Sciences, Istanbul Technical University, Istanbul, Turkey
[7]Wind Energy Unit, CIEMAT, Madrid, Spain
[8]Dept. of Earth Physics and Astrophysics, University Complutense of Madrid, Madrid, Spain
[9]Institute of Numerical Modelling, Department of Physics, University of Latvia, Riga, Latvia
[10]Vestas Wind Systems A/S, Aarhus, Denmark
[11]Wind Energy Department, National Renewable Energy Centre (CENER), Sarriguren, Spain
[*]These authors contributed equally to this work

**Correspondence:** Martin Dörenkämper (martin.doerenkaemper@iwes.fraunhofer.de), Bjarke T. Olsen (btol@dtu.dk)

**Abstract.** This is the second of two papers that document the creation of the New European Wind Atlas (NEWA). In Part 1, we described the sensitivity experiments and accompanying evaluation done to arrive at the final mesoscale model setup used to produce the mesoscale wind atlas. In this paper, Part 2, we document how we made the final wind atlas product, covering both the production of the mesoscale climatology generated with the Weather Research and Forecasting (WRF) model and

5  the microscale climatology generated with the Wind Atlas Analysis and Applications Program (WAsP). The paper includes a detailed description of the technical and practical aspects that went into running the mesoscale simulations and the downscaling using WAsP. We show the main results from the final wind atlas and present a comprehensive evaluation of each component of the NEWA model chain using observations from a large set of tall masts located all over Europe. The added value of the WRF and WAsP downscaling of wind climatologies is evaluated relative to the performance of the driving ERA5 reanalysis and shows that the WRF downscaling reduces the mean wind speed bias and spread relative to that of ERA5 from $-1.50\pm1.30$

10 to $0.02\pm0.78$ $\mathrm{m\,s^{-1}}$. The WAsP downscaling has an added positive impact relative to that of the WRF model in simple terrain. In complex terrain, where the assumptions of the linearised flow model break down, both the mean bias and spread in wind speed are worse than the mesoscale results.





# 1 Introduction

Prior to every new wind turbine and wind farm installation, an energy yield assessment is carried out. This local energy yield assessment is typically based on a combination of wind speed measurements and model data (Rohrig et al., 2019). While the measurements are typically collected at a later stage of the planning phase, model data are used during many stages of the wind

resource assessment. Thus, accurate modelling and evaluation of the modelling compared to observations is important.

The initial evaluations of the wind conditions are typically based on numerical products, such as wind atlases, that generally provide convenient, fast, and easy access to estimations of the characteristic wind conditions at a site. Wind atlases have a long history in wind energy siting applications. In 1989, the European Wind Atlas (EWA, Troen and Petersen, 1989), one of the first comprehensive wind atlases, was published. It documents the meteorological basis for large parts of Europe and was the first

wind atlas to be produced using the *wind atlas method*, a collection of statistical models that are the core of the Wind Atlas Analysis and Application Program (WAsP) software package (Mortensen et al., 2011). EWA was made based on a network of observational masts covering much of Europe, whose measured wind climate served as the input to the WAsP model.

Due to the ongoing advances of numerical weather prediction (NWP) models and the increase in available computational resources, modern wind atlases are typically based on numerical mesoscale model simulations. The wind climatologies (e.g. the

long-term record of wind speed and direction at various levels in the boundary layer) from these simulations, can be downscaled using a microscale model, which can be of different levels of complexity and accuracy depending on the available computational resources and the complexity of the area of interest. However, in regions with homogeneous surface conditions, such as over the sea, over large lakes, grasslands, or deserts, it may be adequate to use the output directly from a mesoscale model to create wind atlases, as e.g. in Peña Diaz et al. (2011) and Doubrawa et al. (2015). In regions with complex terrain, down-

scaling using a microscale model is typically applied. For smaller regions like counties or federate states, statistical-dynamical downscaling based on meso-$\gamma$-scale or microscale Computational Fluid Dynamics (CFD) Reynolds-Avaraged Navier-Stokes (RANS) models can be applied (as in e.g. MWKEL, 2013). For large regions, such as entire countries or continents, linearised flow models, such as WAsP, are most often used to reduce the computational demands (see e.g. GWA, 2019; Mortensen et al., 2014).

Since the start of the century, wind atlases have been produced for a large number of countries, including Egypt (Mortensen et al., 2006), South Africa (Mortensen et al., 2014; Hahmann et al., 2014, 2018), Finland (Tammelin et al., 2013), Germany (Weiter et al., 2019), Greece (Kotroni et al., 2014), Russia (Starkov and Landberg, 2000), Iceland (Nawri et al., 2014), and for offshore regions, such as the Great Lakes in the USA (Doubrawa et al., 2015), the offshore Dutch Wind Atlas (Wijnant et al., 2019), the Southern Baltic (Peña Diaz et al., 2011) and the Southern North Sea (Drüke et al., 2014). A comprehensive

summary of national and regional wind atlases in Europe and can be found in Badger et al. (2018).

With the ongoing technical, computational, and scientific improvements since the release of EWA in 1989, a new updated wind atlas for Europe using current best practices and state-of-the-art models was needed (Petersen et al., 2013). To achieve this, the New European Wind Atlas (NEWA) project was created as a four-year research project with goals to create such a wind atlas, collect relevant field measurements for validation (Mann et al., 2017), and improve the model-chain used for wind climate





downscaling (Sanz Rodrigo et al., 2020). The NEWA wind atlas (Petersen, 2017)[a] consists of mesoscale and microscale datasets that cover all European Union member-states, Norway, Switzerland, the Balkans, and Turkey. The mesoscale atlas was made using the Weather Research and Forecasting (WRF) model (Skamarock et al., 2008) and includes a number of both surface and boundary layer meteorological variables with a $3 \times 3$ km grid-spacing. The resulting data are available at seven wind energy

relevant heights for 30 minute intervals over a 30 year period from 1989 to 2018. The WRF model output was downscaled using the WRF-WAsP methodology (Hahmann et al., 2019) to create the microscale atlas, which is a high-resolution atlas of the statistical wind climate covering the regions in a $50 \times 50$ m grid.

This paper is the second of two parts describing the modelling involved in the making of the NEWA Wind Atlas: the first paper (Hahmann et al., 2020b) deals with the sensitivity experiments that were carried out to guide the selection of the WRF

model configuration used for the production of the mesoscale model simulations of the wind atlas. This paper, the second part, focuses on the production of the wind atlas, including a selection of the results and the evaluation of the wind atlas model-chain using measurements from tall meteorological towers covering most of Europe.

Throughout this paper, we describe the configuration and model adaptations that were used to create the products of the NEWA wind atlas. In addition, issues such as the computational challenges and resources needed for producing a mesoscale

wind atlas for the European continent, and the parallelisation of many millions of WAsP simulations are discussed. Finally, we present an evaluation of the ability of the final NEWA wind atlas to reproduce the spatial variability of the wind at a large number of tall masts in Europe. In Section 2 we introduce the models, the model setups, and discuss the computational aspects of the wind atlas generation. Section 3 presents the main results and the evaluation of the atlas against mast data. In the last part of the paper, we discuss the results (Section 4), provide our conclusions (Section 5), and give a short outlook for potential

future work that may build upon the results presented in this study.

## 2 Wind atlas generation

This section presents the mesoscale and microscale models used in the NEWA model chain to create the wind atlas products. In addition to introducing the models and their setup, the computational, technical, and logistical aspects of both modelling activities are presented.

### 2.1 Mesoscale modelling: the WRF model

#### 2.1.1 The WRF model

The mesoscale wind atlas was created using the WRF model (Skamarock et al., 2008), which has a long record of use for wind energy applications, both on- and offshore (e.g. Storm et al., 2009; Jimenez et al., 2010; Liu et al., 2011; Horvath et al., 2012; Karagali et al., 2013; Hahmann et al., 2015; Draxl et al., 2015; Dörenkämper et al., 2015; Lundquist et al., 2019). The setup of

---

[a]https://map.neweuropeanwindatlas.eu



the WRF model used for the NEWA wind atlas is based on the evaluation of a large number of sensitivity experiments using mast data, which is documented in the first part of this study (Hahmann et al., 2020b).

The mesoscale model simulations of the NEWA production run use a modified version of the WRF model version 3.8.1, with changes in the Mellor-Yamada-Nakanishi-Niino (MYNN) planetary boundary layer (PBL) scheme (for details see Section 5.2 in Hahmann et al. (2020b)). Furthermore, additional code that estimates ice accumulation was added to the WRF model code. This icing model is based on the ice growth model from Makkonen (2000).

The 30-year mesoscale database is created by running a series of WRF model simulations: seven days plus a 24 h spin-up period, which overlaps with the last day of the preceding weekly run. These relatively long simulations guarantee that the mesoscale flow is in full equilibrium with the mesoscale aerodynamic characteristics of the terrain, while the nudging is used to keep the model solution from drifting away from the observed large-scale atmospheric patterns (Vincent and Hahmann, 2015). An advantage of the weekly runs is that the simulations are independent of each other and can be integrated in parallel. This reduces the total wall clock time needed to complete a multi-year climatology at a decent computational overhead. However, the state of the lower boundary, which is in equilibrium with the PBL conditions, is lost after each re-initialisation, necessitating the 24 hours of spin-up time.

All mesoscale simulations used three nested domains with a 3 km horizontal grid spacing for the innermost grid and a 1:3 ratio between inner and outer domain resolution, leading to 3 different resolutions: 27 km for the outer domain (D1), and 9 km and 3 km for the inner nested domains D2 and D3. The area to be covered by the NEWA wind atlas was divided into ten independent high-resolution computational domains (named BA, CE, FR, GB, GR, IB, IT, SA, SB and TR) as shown in Fig. 1. These are the inner-most domains (D3), which all share the common outer-most domain (D1). However, to further parallelize the simulations, each inner-most domain was run separately from the others. The ten domains were created using the following rules:

1. Domains have to cover the NEWA area of interest: all European Union member-states, Norway, Switzerland, the Balkans, and Turkey, offshore areas 100 km off each coast, complete North and Baltic Seas.

2. Domains should not include large regions outside the NEWA area of interest.

3. Domains must be large enough so that each country is fully covered in one domain (exception: Norway, Sweden and Finland).

4. Domains must have sufficient overlap: at least 30 grid points buffer at each domain boundary. (following Wang et al., 2019).

The final domains vary in size from 325 × 343 grid points (GR) to 631 × 415 grid points (SB). All D2 domains have a blending zone of 350 km (or 39 grid squares) around the respective D3 domain. The common outer D1 domain is 250 × 220 grid points, and corresponds to the background map shown in Fig. 1.





**Table 1.** Setup configuration used in the NEWA production run.

| | |
|---|---|
| WRF version | 3.8.1 (modified PBL + icing code)[a] |
| Domains | 10 domains (see Fig. 1); Lambert conformal map projection |
| Grid spacing ($\Delta x, \Delta y$) | 3 nests: 27 km (D1), 9 km (D2), 3 km (D3); 1 way nesting |
| Vertical discretisation | 61 vertical levels, model top at 50 hPa |
| Model levels | 20 model levels below 1 km |
| | 10 lowest level heights: approx. 6, 22, 40, 57, 73, 91, 113, 140, 171, 205 m a.g.l. |
| Simulation length | 8 days including 24 h spin-up |
| Terrain Data | Global Multi-resolution Terrain Elevation Data 2010 at 30" (Danielson and Gesch, 2011) |
| Land use data | CORINE 100 m (Copernicus Land Monitoring Service, 2019), |
| | ESA CCI (Poulter et al., 2015) where CORINE not available |
| Dynamical forcing | ERA5 (Hersbach and Dick, 2016) reanalysis ($0.3° \times 0.3°$ resolution) on pressure levels |
| Sea conditions | OSTIA citepostia SST and sea-ice (1/20°, approx. 5 km) |
| Lake temperature | average ground temperature from ERA5, |
| | lakes are converted to ocean when temperature is present in OSTIA |
| Nudging | Spectral nudging in D1 only, above PBL and level 20 |
| Time step | adaptive ($<5\%$ failed) |
| PBL | MYNN (modified) (Mellor and Yamada, 1982) |
| Surface layer | MO (Eta similarity) (Janjic and Zavisa, 1994) |
| Land surface model | NOAH-LSM (Tewari et al., 2004) |
| Cloud microphysics | WRF Single-Moment 5-class scheme (Hong et al., 2004) |
| Radiation | RRTMG scheme, 12 min calling frequency (Iacono et al., 2008) |
| Cumulus parameterisation | Kain-Fritsch scheme on D1 and D2 (Kain, 2004) |
| Icing | WSM5 (Hong et al., 2004) + icing code + sum of qcloud and qice |
| Diffusion | simple diffusion |
| | 2D deformation |
| | 6th order positive definite numerical diffusion |
| | rates of 0.06, 0.08, and 0.1 for D1, D2, and D3 |
| | vertical damping |
| Advection | positive definite advection of moisture and scalars |
| Numerical options | 480 cores, IO Quilting (1 node used for output) |

[a] The WRF code modifications are available from the NEWA GitHub repository: https://github.com/newa-wind/Mesoscale.

All other final WRF model setup parameters of the NEWA production runs are summarised in Table 1. The above mentioned WRF model code modifications as well as the namelists and domain files for all ten domains are available from the NEWA GitHub repository (https://github.com/newa-wind/Mesoscale).



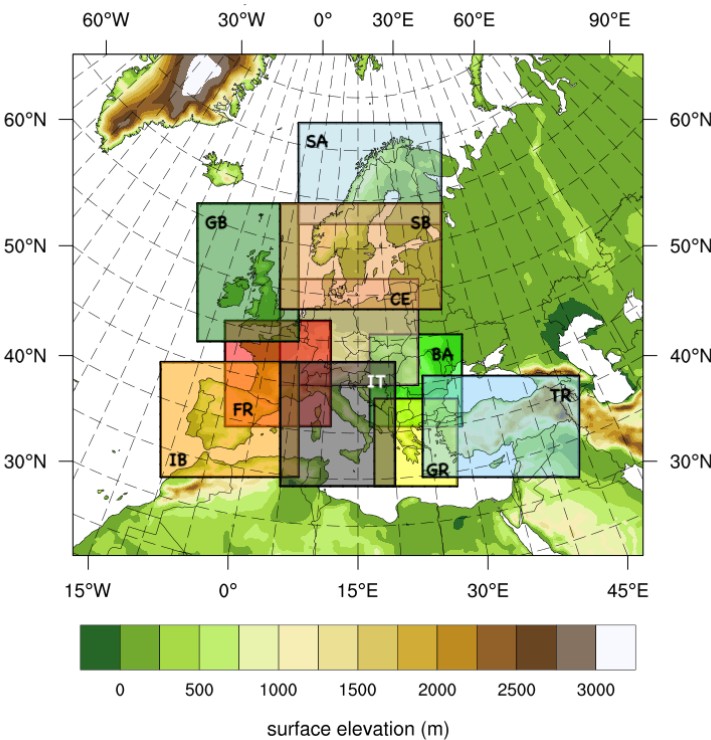

**Figure 1.** The location of ten WRF model domains (D3) used in the NEWA production run, excluding a 30 grid points buffer around each domain. The background map corresponds to D1 which is the same for all simulations. D2 domains are not shown (reproduced from Hasager et al., 2019)

.

### 2.1.2 Computational aspects

The production run for the NEWA mesoscale was conducted between August 2018 and March 2019 on the MareNostrum4 supercomputer that is operated by the Barcelona Supercomputing Center (BSC). For this purpose, computational resources to the amount of 57 million core hours were granted to the consortium via a PRACE (Partnership for Advanced Computing in Europe) proposal. The PRACE project was active and the resources were available between April 2018 and March 2019. Besides the computational resources in terms of core hours, 100 terabyte (TB) of scratch space (for the temporary files of the runs) and an additional 100 TB of project space (for storing e.g. reanalysis input data and scripts) was available on the system.

As described in Section 2.1.1, the final setup for the production run was divided into ten domains covering the EU plus Norway, Switzerland and Turkey. The estimated total mesoscale model output *raw* data (10 domains over 30 years) in 3 km spatial and 30 min temporal resolution would have resulted in a total of 6 petabyte. However, the final post-processed wind atlas however resulted in a much lower volume of around 0.2 petabyte (see below). Consequently, a partitioning of the full run





into smaller runs that fit into the 100 TB of scratch space was essential, and so was a high degree of automation of the runs, the post-processing and the data transfer.

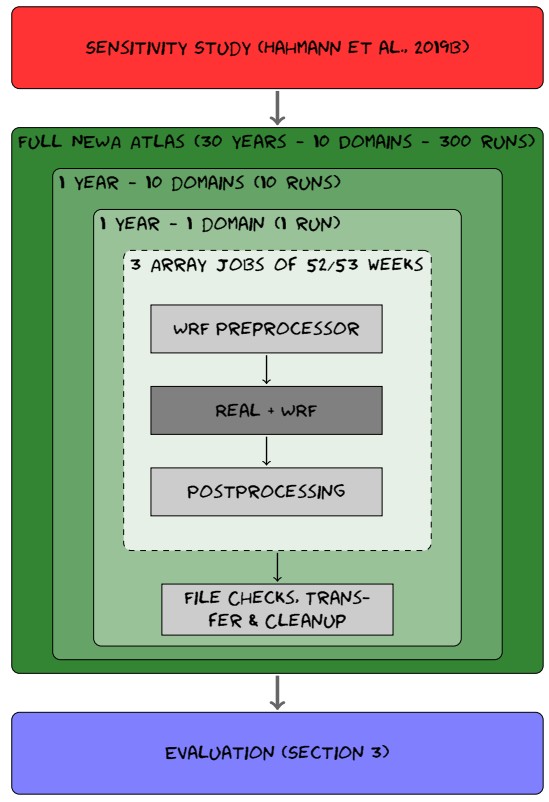

**Figure 2.** Illustration of how the production run was conducted. The dashed line encloses fully automatic processes, the dark grey box the parallel process (here 480 cores) and light grey serial processes.

Figure 2 illustrates the split of the full wind atlas into separate computational tasks and the degree of automation of each part of the job chain. The full 30 year wind atlas for all 10 domains was first subdivided by the year and then by the domain. Each

5 year-domain run (300 runs in total) then contained three array jobs of 52 or 53 elements each, one for each of the weekly runs. Out of these three array jobs, only the execution of the mesoscale run itself (`real.exe` and `wrf.exe` from WRFV3.8.1; indicated by the dark grey colour in Figure 2) were run in parallel, using 480 cores each. All other tasks were run in serial (light grey colour in Figure 2).

The setup and submission of each year-domain set (three array jobs of 52/53 weeks each) was automated using a python3

10 script adapted to the properties of the computing cluster. It linked, copied, and adapted the necessary input files (e.g. namelists) and then submitted the job scripts to the queuing system. Dependencies were setup between each of the three stages of the job arrays to allow a full automation of the simulation process. This means that the array of jobs responsible for post-processing



automatically started after the main runs were completed. In case of problems (e.g. hardware issues), a different script was used to manually re-submit single weeks of a year-domain run.

In total each year-domain run occupied about 20 TB (from 13 TB for the GR domain to 23 TB for the SB domain) of scratch space including all raw and post-processed data. Thus, it was possible to have up to five year-domain sets running at the same

time within the space provided on MareNostrum 4. After a year-domain set completed successfully (i.e. the post-processing array run completed), a script for file checks, transfer, and cleanup was started. The file checks portion of the script checked the post-processed output files for consistency and completeness. If the check proved successful, the post-processed files were moved to a special transfer directory and the raw data was deleted. On a dedicated server at the Technical University of Denmark (DTU), a cron job was constantly watching for post-processed files in the aforementioned transfer directory and

initiated the transfer to the DTU server when new post-processed files were found.

During the production run, between August 2018 and March 2019, the mesoscale working group of the NEWA consortium (the authors of this study and further supporters) was constantly on standby. Each week a different person was on duty to launch, check, resubmit, and transfer runs seven days a week to ensure fast progress and to avoid longer idle periods.

In terms of computational costs, each year-domain configuration spent 80,000–140,000 core hours, leading to a total of

about 35 million core hours for the full wind atlas. (The remaining PRACE grant was used for ensemble run calculations, see e.g. González-Rouco et al. (2019)). The resulting post-processed mesoscale time series (daily netCDF-files following CF-1.6 conventions) contained the parameters given in Appendix A. The mesoscale wind atlas (30 years, 10 domains, 30 min resolution, 7 vertical levels) resulted in a total volume of around 160 TB.

The final NEWA mesoscale wind atlas was created by combining the results from the individual mesoscale domains into a

single merged mesoscale dataset for public use. Figure 3 shows how each domain contributed to the combined domain. Because all domains shared the same outer domain, reference location, and projection, the grid nodes of neighbouring domains overlap exactly, so no interpolation was needed to combine them. Whenever possible, data for each country's exclusive economic zone comes from the same mesoscale domain (cf. Figure 1).

## 2.2   Microscale modelling

### 2.2.1   The WRF-WAsP methodology

The effective horizontal resolution of the mesoscale atlas is several kilometers (Skamarock, 2004) and cannot capture local flow features from sub-grid variations in orography and surface roughness. However, capturing these effects can be vital for accurately determining the local wind climate at a site (Sanz Rodrigo et al., 2017). Therefore, the WRF model output was further downscaled using the linearised microscale model WAsP (Troen and Petersen, 1989) following the WRF-WAsP

downscaling methodology (Hahmann et al., 2014; Badger et al., 2014), to simulate the effect of the local topography on the wind climate. WAsP is a collection of linear models for downscaling, which includes sub-models for the effects of: orography, surface roughness, obstacles, atmospheric stability, and vertical extrapolation (Troen and Petersen, 1989). These models enable horizontal and vertical extrapolation from an observed or modelled wind climate at one site to other, nearby sites. In WAsP the

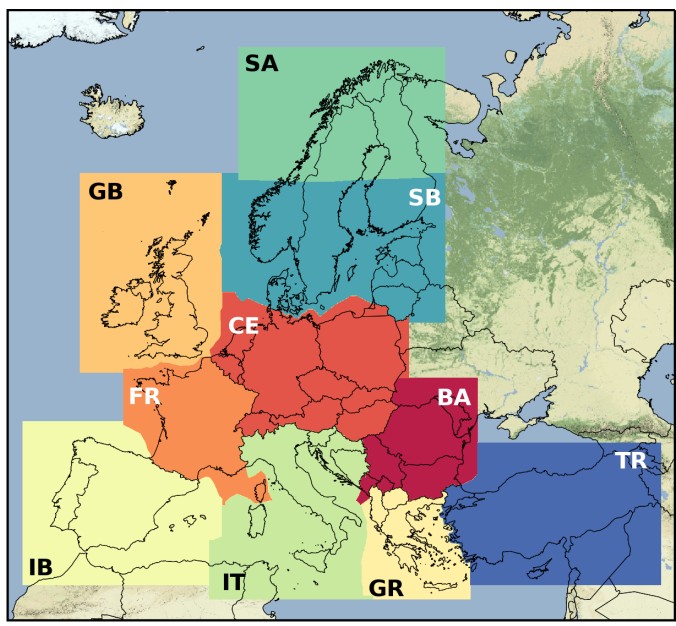

**Figure 3.** Map showing how the individual mesoscale domains are combined into a single merged dataset. Whenever possible the data for each country's exclusive economic zone comes from the same domain. The background is the stamen terrain-background from http://maps.stamen.com/terrain-background - © OpenStreetMap contributors 2020. Distributed under a Creative Commons BY-SA License.

wind climate at a given location is defined as the probability density of wind speed and wind directions. It can be represented as the probability density binned into a number of equal-width wind speed and wind direction bins or as a set of frequencies and best fit Weibull distribution parameters for each sector.

The typical WRF-WAsP procedure involves two steps. First, the WRF model wind climate is "generalised", which removes
the WRF local terrain effects from the WRF-simulated wind climate (i.e. the wind speed and sector statistics) to produce a new wind climate that is representative of a larger area surrounding the model grid cell. The *generalised wind climate* (in WAsP terminology) corresponds to the wind field distribution that would exist without orography and a homogeneous surface roughness, i.e. a flat surface of constant surface roughness. Therefore, the generalised wind climate varies only with height. In the WRF-WAsP process, each generalised wind climate holds information, not just of the wind field distribution for a single
surface roughness and height above ground level (a.g.l.), but for a number of pre-selected surface roughnesses and heights. In the second step, local terrain effects are "applied" to the generalised wind climate, resulting in a *predicted wind climate*. The local terrain effects vary based on the site location and height above the surface. The generalisation procedure is carried out for all WRF model grid points, and the prediction procedure is carried out at each microscale model grid-point and desired height. Resulting in a high-resolution map of the wind climate, such as the one presented in the right panel of Fig. 6. The default
procedure of WAsP is to represent the wind climate as binned wind speeds and directions during the generalisation step until all corrections have been applied, then Weibull fits are made, and used in subsequent steps.





In the WRF-WAsP methodology, the terrain effects for the generalisation are derived from the gridded WRF model elevation and surface roughness, while the terrain effects used for prediction are estimated using the best available high-resolution maps of elevation and surface roughness. The detailed technique for how these effects are calculated and used is described in Badger et al. (2014) for the mesoscale generalisation and Troen and Petersen (1989) for the microscale prediction. All effects are taken into account using the default treatment of atmospheric stability in WAsP, which over land corresponds to an average heat flux of $-40\,\mathrm{W\,m^{-2}}$ and root-mean-square of $100\,\mathrm{W\,m^{-2}}$ and over sea an average heat flux of $15\,\mathrm{W\,m^{-2}}$ and root-mean-square of $30\,\mathrm{W\,m^{-2}}$.

In NEWA, the long-term wind atlas, based on the 30 years of WRF model data, was made using the default WRF-WAsP downscaling method described above. The WRF model wind climates from these 30 years were generalised to heights of 50, 75, 100, 150, and 200 m a.g.l. and to surface roughnesses of 0.0002, 0.03, 0.1, 0.4, and 1.5 m. Subsequently, predictions were made at heights of 50, 100, and 200 m a.g.l. on a 50 m × 50 m horizontal grid across Europe.

### 2.2.2 High-resolution surface datasets

The high resolution terrain elevation was created by combining Shuttle Radar Topography Coverage Version 3 (SRTM v3) dataset (Farr et al., 2007) south of 60 °N and the ViewFinder DEM (de Ferranti, 2014) north of 60 °N. Both of these datasets have a 3 arc-second ($\approx 90\,\mathrm{m}$) resolution and are provided in the WGS84 map projection. The high resolution surface roughness length ($z_0$) values were created based on the 2018 CORINE land cover dataset (Copernicus Land Monitoring Service, 2019), which has a horizontal resolution of 100m and is in the ETRS89 Lambert Azimuthal Equal-Area map projection (EPSG 3035).

For downscaling with WAsP, the CORINE land cover classes were related to constant $z_0$ values through a look-up table (Table 2, "WAsP" column). Since no objective or thoroughly validated land-use to surface roughness conversion exists for the CORINE land-use classes, the accuracy of the table is highly uncertain. For WRF, the 44 CORINE land-use classes were converted to the most similar 21 class USGS (Anderson et al., 1976) and converted to surface roughness length using constant values first suggested by Pineda et al. (2002) (see Table B1 in Appendix B and Hahmann et al. (2020b) for further details). Silva et al. (2007) proposed surface roughness values for the CORINE land-use classes, but these were only validated for three sites in Portugal. A different conversion table, referred to as *DTU Table*, was proposed (private communication, R. Floors, N. G. Mortensen, and A. Hahmann, DTU Wind Energy, March 2019), but likewise has not been comprehensively validated.

The $z_0$ values in Table 2 "WAsP" column, were determined by using the DTU table as a starting point, and then adjusting the values toward the corresponding $z_0$ in the WRF model table (Table 2, "WRF" column). Since the values in the DTU table are defined for the 44 CORINE classes, each class has a better characterisation of a specific land use type. In contrast, each of the 21 USGS classes needs to represent a broader range of land-use types. This means that the DTU table has more variation than the USGS table, and in most cases low roughness land-use types have much lower $z_0$ values and high roughness land-use types much higher ones. The WAsP $z_0$ values were adjusted to be more like the WRF values to reduce the difference in the effective surface roughness between the two models, and therefore, reduce the risk of over correcting the wind climate when using the microscale model. For example, "non-irrigated arable land" and "pastures" are common land-use classes in Europe (Table 2, "Proportion" column), and are respectively assigned roughnesses of 0.05 m and 0.03 m in the DTU table (not shown). However,





in the WRF vegetation table, they are both assigned a value of 0.10 m, which — everything else being equal — results in a large increase in the predicted wind speed when downscaling the WRF data for these classes. Similarly, locations with high $z_0$ values, such as forests and urban areas, experience the opposite effect. Therefore, the adjustment towards the WRF roughness values can be viewed as making the WAsP roughness corrections more conservative.

**Table 2.** Surface roughness length [m] for each land-use category in WAsP and WRF and the proportion it represents in the total dataset.

| Category | Proportion [%] | WAsP $z_0$ [m] | WRF $z_0$ [m] | Category | Proportion [%] | WAsP $z_0$ [m] | WRF $z_0$ [m] |
|---|---|---|---|---|---|---|---|
| Continuous urban fabric | 0.1 | 1.0 | 1.0 | Broad-leaved forest | 8.0 | 1.0 | 0.9 |
| Discontinuous urban fabric | 2.3 | 1.0 | 1.0 | Coniferous forest | 11.1 | 1.2 | 0.9 |
| Industrial or commercial units | 0.4 | 0.7 | 0.5 | Mixed forest | 4.2 | 1.1 | 0.5 |
| Road and rail networks and assoc. land | 0.1 | 0.2 | 0.5 | Natural grasslands | 2.9 | 0.1 | 0.1 |
| Port areas | < 0.1 | 0.5 | 0.5 | Moors and heathland | 2.4 | 0.12 | 0.12 |
| Airports | < 0.1 | 0.1 | 0.5 | Sclerophyllous vegetation | 1.5 | 0.12 | 0.12 |
| Mineral extraction sites | 0.1 | 0.15 | 0.5 | Transitional woodland-shrub | 4.1 | 0.4 | 0.12 |
| Dump sites | < 0.1 | 0.15 | 0.5 | Beaches - dunes - sands | 0.1 | 0.01 | 0.01 |
| Construction sites | < 0.1 | 0.3 | 0.5 | Bare rocks | 1.3 | 0.05 | 0.01 |
| Green urban areas | < 0.1 | 0.8 | 0.5 | Sparsely vegetated areas | 3.2 | 0.03 | 0.01 |
| Sport and leisure facilities | 0.2 | 0.3 | 0.5 | Burnt areas | < 0.1 | 0.2 | 0.01 |
| Non-irrigated arable land | 16.5 | 0.1 | 0.1 | Glaciers and perpetual snow | 0.2 | 0.005 | 0.001 |
| Permanently irrigated land | 1.5 | 0.1 | 0.1 | Inland marshes | 0.2 | 0.05 | 0.001 |
| Rice fields | 0.1 | 0.1 | 0.1 | Peat bogs | 1.6 | 0.03 | 0.001 |
| Vineyards | 0.6 | 0.3 | 0.2 | Salt marshes | 0.1 | 0.02 | 0.001 |
| Fruit trees and berry plantations | 0.6 | 0.4 | 0.2 | Salines | < 0.1 | 0.005 | 0.001 |
| Olive groves | 0.7 | 0.4 | 0.2 | Intertidal flats | 0.2 | 0.001 | 0.001 |
| Pastures | 5.7 | 0.1 | 0.1 | Water courses | 0.2 | 0.0002 | 0.0001 |
| Annual crops assoc. with perm. crops | 0.1 | 0.2 | 0.2 | Water bodies | 1.8 | 0.0002 | 0.0001 |
| Complex cultivation patterns | 3.3 | 0.2 | 0.2 | Coastal lagoons | 0.1 | 0.0002 | 0.0001 |
| Agriculture with sig. areas of nat. veg. | 3.7 | 0.2 | 0.2 | Estuaries | 0.1 | 0.0002 | 0.0001 |
| Agro-forestry areas | 0.5 | 0.5 | 0.2 | Sea and ocean | 20.1 | 0.0002 | 0.0001 |

### 2.2.3 Computational aspects

For easy interfacing with the WAsP model the "PyWAsP" (PyWAsP, 2020) software package developed at DTU was used. PyWAsP is a python wrapper around the (mostly) Fortran-based WAsP core. The WAsP calculations are independent, making





the downscaling procedure an "embarrassingly parallel" problem. However, to facilitate the 5.6 billion WAsP calculations that needed to be done, a two-tier parallelisation process was employed, taking advantage of the cluster architecture used.

First, to split the work into manageable chunks, the wind atlas area was divided into 1402 tiles, each covering an area of $100 \times 100 \, \text{km}$, and thereby consisting of $2000 \times 2000$ calculation points (the target locations) spaced 50 m apart, adding up to the
5.6 billion calculations mentioned above. The tiles and all WAsP modelling was defined in the ETRS89 Lambert Azimuthal Equal-Area map projection (EPSG 3035) in metric units of metres. Each tile was submitted to a single computational node, with the load balancing of tile jobs managed by the HPC workload manager. To take advantage of the 32 CPU cores on each compute node, each tile was divided into 2500 sub-tiles of $40 \times 40$ calculation points. The *dask* python package was used for scheduling and distribution of the computations needed for each of the sub-tiles across the different CPU cores.

To allow for each tile to operate as a stand-alone computational task, terrain and generalised wind climate data was created for each tile in advance. Natural neighbours interpolation (Sibson, 1981) was used to interpolate the generalised wind climates computed from the mesoscale model output to each target location. To ensure that a sufficient number of input wind climates were available around each target location for the interpolation, the wind climates included a 10 km buffer region around each tile.

Following WAsP best practices (Mortensen, 2018), a buffer area of 25 km around the tile should be enough to accurately model the influence of the orographic and surface roughness maps. However, this assumes some human judgement in the creation of the surface roughness map. WAsP uses the $z_0$ map in a couple of different ways. First a roughness rose is created. It contains information about upstream roughness changes in a number of radial sectors (typically 12) and its distances to the point. For computational efficiency only the roughness changes that most impact the flow is kept (ten at most). These are
identified based on the amount of total roughness variation they account for. The roughness rose is then used to calculate the upstream "background" roughness for the point, and to model the internal boundary layers caused by the roughness changes. One of the key limitations of this approach is that the model assumes the last roughness in the roughness rose continues to be the roughness indefinitely. Therefore, when creating a roughness map, it is important to look further upstream when making the map and to ensure that the outer roughness values match those found further upstream of the site. However, in our automated
process this is not possible. This led to sensitivities in the sub-tile results, due to the inclusion of additional upstream roughness values when calculating the background roughness. To limit this impact, we included a preprocessing step that calculated the background roughness at a 1 km grid spacing using roughness maps that extended 100 km upstream from the grid point. When calculating the predicted wind climate, a 25 km map was used, but the preprocessed background roughness from a point approximately 30 km upstream was inserted into the roughness rose as the last roughness to reduce the sub-tile sensitivity.

In summary, to provide the correct background roughness to the WAsP model, the following steps were carried out, for each sub-tile: 1) get coarse background roughness from pre-processed map, 2) create roughness rose, 3) add coarse background roughness as last bin of roughness rose 4) calculate roughness site effects using the updated roughness rose. The computational time for the simulation of the tiles varied considerably, between 1–14 h for each tile, depending on the complexity of the terrain.





## 2.3 Evaluation methodology

It is not possible or appropriate to evaluate the wind climatology of the wind atlas itself (e.g. what is available for download from the NEWA site) because it represents a long climatological period (1989–2018) and no wind speed measurements span that entire period at wind energy relevant heights. Instead, the NEWA model-chain is validated by using it to create individual wind

climates for 291 tall masts covering Europe, such that the wind climates represent the exact measurement periods covered by each mast. This section covers the metrics used to describe the terrain complexity at each site, followed by a description of the masts' measurement data. Finally, we describe a number of modifications to the WRF-WAsP methodology made specifically for the evaluation against mast data.

### 2.3.1 Terrain complexity

To quantify the relationship between model biases and the complexity of the terrain at the sites, several metrics related to the orographic and surface roughness complexity were calculated for each site. In this paper we focus on the ruggedness index (RIX) (Mortensen et al., 2008), which was used to quantify the orographic complexity. The RIX number is defined as the fractional extent of the terrain that exceeds a critical slope, in this case $16.7\,^\circ$, within $3500\,\mathrm{m}$ of the point of interest. The RIX number is used in WAsP to indicate terrain where the surrounding orographic slopes are steeper than the valid limits of the

flow model (IBZ Jackson and Hunt, 1975) and thus where the orographic speed-ups are expected to be overestimated.

Additionally, three metrics to quantify the surface roughness heterogeneity were investigated for each site. First, the degree of variation of the surface roughness around the sites was used to identify sites that would likely have complex structures due to rapid changes in the surrounding surface roughness. Second, the distance from the mast location to the nearest coastline was used to detect coastal influences on the model biases. Third, the average aggregate upstream surface roughness at the site was

used to detect biases associated with high or low roughness sites.

Initial analysis showed that each of the surface roughness metrics explained some of the variance of the model biases. However, it was clear that the RIX number explained most of the variance for both the ERA5 and WAsP results, and a large amount of the variance in the WRF model results. Therefore, only the RIX number is included in the remaining analysis.

### 2.3.2 Observed data

The results of the NEWA model-chain were validated against measurements from 291 tall masts made available for the study. Because the data is proprietary, only aggregated results are presented. Figure 4 shows the number of masts located in each country. Large variations in the number of masts present in each country and across regions exist, e.g. just 4 masts in Germany and Spain, while 38, 42, and 44 masts are located in Poland, Italy, and Turkey respectively. However, most parts of Europe and Turkey are well represented.

Figure 5 shows the distribution of several descriptive variables for the 291 masts. All measurements used in the evaluation were taken on tall meteorological masts. Cup or sonic anemometers were used for the wind speed measurements, either from a single instrument or via an optimal sampling of measurements from two instruments mounted on opposing booms to reduce

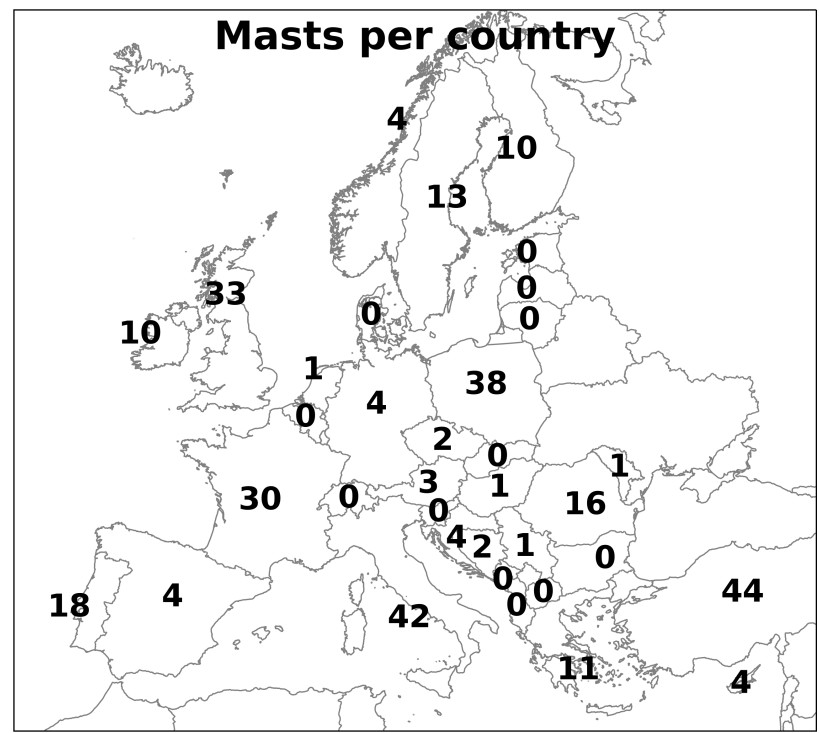

**Figure 4.** The number of masts located in each country in Europe. For several smaller countries the number has been omitted for readability (all of them have zero masts).

flow distortion effects. Only measurements 40–150 m a.g.l. were used for the evaluation (Fig. 5b) in order to avoid the large uncertainties associated with wind speed measurements near the surface, and to ensure that the measurements are representative of modern and future turbine hub heights. Wind direction measurements were taken either from the sonic anemometers or from wind vanes as close to the wind speed measurements as possible (typically 0–40 m below the wind speed instrument).

5 The measurements were previously quality controlled by applying an in-house method of the data provider and were further checked for obvious measurement errors like icing and non-physical (e.g. repeated) signals for this study; fortunately no problems were detected.

Twelve months of measurements were used from each mast to ensure that the results were not biased due to differences in sample sizes. The period with the highest availability of measurements, between 2007 and 2015, was chosen. For most

10 masts, the best period occurred after 2009 (Fig. 5a). To avoid biases due to seasonal variations in data availability, an additional requirement that at least 80 % of the possible data was available for each month was used. For most masts, more than 90 % of the possible data was recovered every month.

The RIX values for the masts (Fig. 5c) shows a skewed distribution, with most values found below 2 %, and only a few masts with very large values. For further analysis, the masts were grouped into three RIX groups: *low*: 0 % ($n = 110$), *medium*:



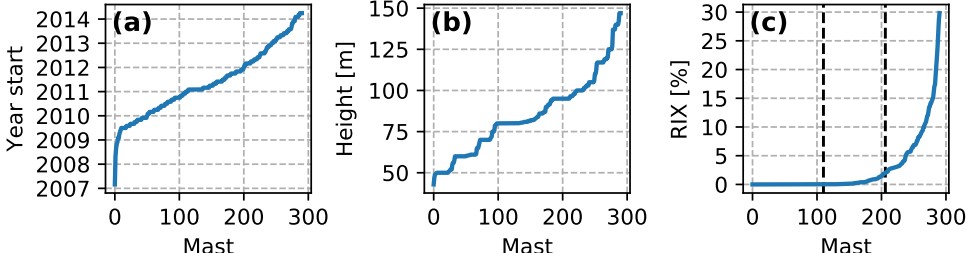

**Figure 5.** Ranked values of metadata variables for the 291 masts. (a) the start-year of the one-year period, (b) the height a.g.l. of the wind speed measurements, and (c) the ruggedness index (RIX). The dashed vertical lines in (c) shows the separation (greater than 0 % and greater than 2 %) of the masts into groups by RIX.

0–2 % ($n = 96$), and *high*: 2 % ($n = 85$) or greater (dashed lines in the figure). These thresholds were chosen to ensure a similar number of masts in each group.

### 2.3.3 Adjustments to WRF-WAsP downscaling used for evaluation against mast data

For evaluation of the downscaled wind climate at each mast site, some modifications to the WRF-WAsP methodology were

5 made. First, the measurements and the WRF model data for the mast location (nearest grid cell) was obtained, ensuring that the WRF data was concurrent to the measurements. Second, instead of a two-step process (generalisation and prediction) with Weibull-fitting used after the generalisation step, the corrections due to terrain effects from both steps were applied to the binned wind climate during the same, single, step. Third, since the WRF output was interpolated to 50, 75, 100, 150, and 200 m, we used the height closest to the height of the measurements, thus the largest vertical extrapolation required was

10 less than 25 m. The probability density of the original bin is distributed to the nearest bins in the new binned wind climate. After repeating this for every bin of the original wind climate, the new predicted binned wind climate is complete. For all effects, neutral atmospheric stability was assumed. This alternative approach has some advantages over the default approach: a Weibull distributed wind is not assumed, so no parameterisation biases occur. This is a particular advantage for shorter periods, i.e. months, but may also be advantageous for one year periods, such as those used here.

Although minimal vertical extrapolation was performed, the assumption of neutral stratification may cause small discrepancies between the approaches used for calculating the long-term wind climates and for the evaluation against mast data. However, A direct comparison between the WRF-WAsP downscaling over the entire map and that at the sites is not possible, but the differences should be in the order of a few percent. This issue is discussed further in Sect. 4.



**Figure 6.** (a) ERA5, (b) WRF, and (c) WAsP mean wind speed at 100 m a.g.l. averaged over the full 30 year period (1989–2018). The zoom-in shows the results for the island of Crete. Additional details can be seen on the NEWA wind atlas website https://map.neweuropeanwindatlas. eu/. The map background is the stamen terrain-background from http://maps.stamen.com/terrain-background - © OpenStreetMap contributors 2020. Distributed under a Creative Commons BY-SA License.





## 3 Results

### 3.1 The wind atlas

Figure 6 shows one of the main results of the NEWA wind atlas, the map of the wind speed at 100 m averaged over the full 30 year period (1989–2018) derived from the mesoscale simulations and downscaling with WAsP. The difference in the wind

speed between onshore and offshore sites is evident in many areas. Local wind systems like the Mistral, the Bora and the flow through the Strait of Gibraltar (Levante and Poniente) are clearly visible. Due to the limited resolution, some of these flows are not fully resolved in commonly used reanalysis data sets like ERA5 (Hersbach and Dick, 2016) and CFSRv2 (Saha et al., 2014) or MERRA2 (Gelaro et al., 2017) (not shown here). The inset figures in Fig. 6 show the flow around the Greek island Crete, which is heavily influenced by the Etesian winds. ERA5 and WRF both capture these winds, but differ significantly in

magnitude over and in the lee of Crete. It is also clear that additional flow-features have been resolved by WRF, e.g. gap-flows in the mountains of Crete. The microscale downscaling with WAsP adds additional details especially over the complex coastal mountain ridges. Additional details and a comparison with satellite data for this local wind system are provided in Hasager et al. (2019).

Additional terrain effects along large mountain ridges can be seen, including the highest wind speeds in Europe, occurring in

Central Norway, and areas in the Alps that have wind speeds above $10 \, \mathrm{m \, s^{-1}}$. As expected, the microscale downscaling results in larger flow variations in mountainous areas, as exemplified in the inset figure. Thus, the wind speeds on the mountain tops are slightly higher.

Figure 7 shows the differences in the 100 m mean wind speed between WRF and ERA5 (a) and WAsP and WRF (b) for the full 30 year period. To calculate the differences the lower resolution data was interpolated bi-linearly to the grid of the

higher resolution dataset, which is not typically recommended, but is done here for illustration purposes. For convenience, the difference between WAsP and WRF was not calculated on the native 50 m WAsP grid, but a down-sampled grid corresponding to every tenth point in north and east directions, i.e. with a $500 \times 500$ m spacing. In general, it is clearly visible that the mesoscale model resolves the terrain better than the reanalysis and thus capture more of the variation due to orography, shown e.g. by the larger wind speed on top of mountain ridges. On the large European scale however, the differences between meso-

and microscale are small, especially in areas of low terrain complexity. However, over very complex topography (e.g. in the Alps and Pyrenees) the WAsP downscaling greatly increases the wind speed. This is studied in more detail in Sect. 3.2.

The post-processed mesoscale and microscale fields can be accessed interactively on the NEWA website: https://map. neweuropeanwindatlas.eu/.

### 3.2 NEWA model-chain evaluation

The evaluation of the NEWA model-chain performed by comparing the wind climates estimated at each stage of the model-chain: ERA5 (forcing reanalysis), ERA5+WRF (mesoscale, simply labelled "WRF") and ERA5+WRF+WAsP (microscale, simply labelled "WAsP") to the observed wind climates. By comparing the wind climates, it is not possible to evaluate the



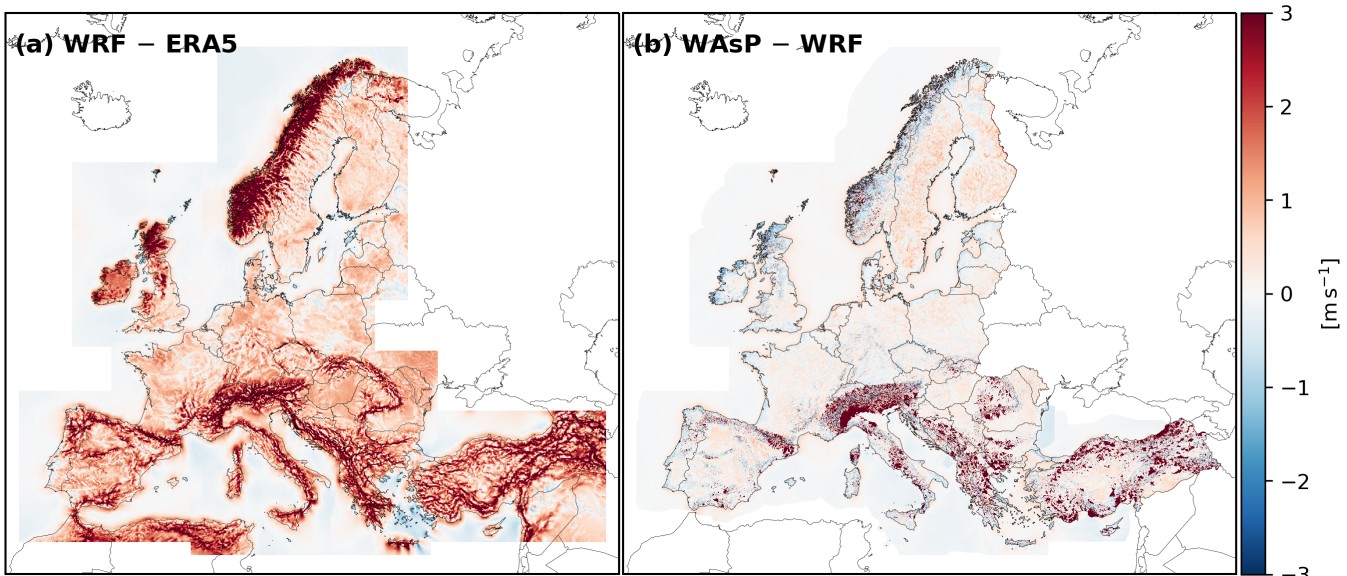

**Figure 7.** Mean wind speed differences for: WRF minus ERA5 (a) , and WAsP minus WRF (b) at 100 m a.g.l. averaged over the full 30 year period (1989–2018). Note that the lower resolution data was interpolated bi-linearly to the grid of the dataset with higher resolution. The difference between WAsP- and WRF-derived winds was calculated on a down-sampled version of the WAsP grid with a grid spacing of 500 × 500 m

time-dependent aspects of the NEWA model results, which, additionally, are not available from the WAsP model. Further analysis of time-dependent aspects are included in the NEWA uncertainty report (González-Rouco et al., 2019).

The WRF- and ERA5-derived wind climates are calculated from the time series of wind speed and direction, and interpolated linearly in time and space to the mast location and height for times concurrent with the measurements. The WAsP wind climates were estimated using the method outlined in Section 2.2.

### 3.2.1 Mean wind speed biases

The statistics of the comparison between observed and modelled mean wind speeds for the 291 masts are presented in Fig. 8. The same point scatter is repeated in three subplots, each with samples from one of the three *RIX* groups highlighted with colours and the remaining samples greyed out (Fig. 8). The corresponding distributions of mean wind speed biases, computed as the model minus the observation values, are shown in Fig. 9. The smallest spread, and least scatter, in mean wind speed, from all three downscaling stages (ERA5, WRF and WAsP), is at *low* RIX sites, and is considerably larger at *medium* and *high* RIX sites. This result is not unexpected and shows that the uncertainty in all three models increases as the orographic complexity of the site increases.

The overall mean wind speed bias for all the masts is $-1.5 \, \mathrm{m\,s^{-1}}$ for ERA5, while it is $0.28 \, \mathrm{m\,s^{-1}}$ and virtually zero for the WAsP and WRF wind speeds, respectively (Fig. 9). The sample means of the RIX groups show that the biases of the ERA5



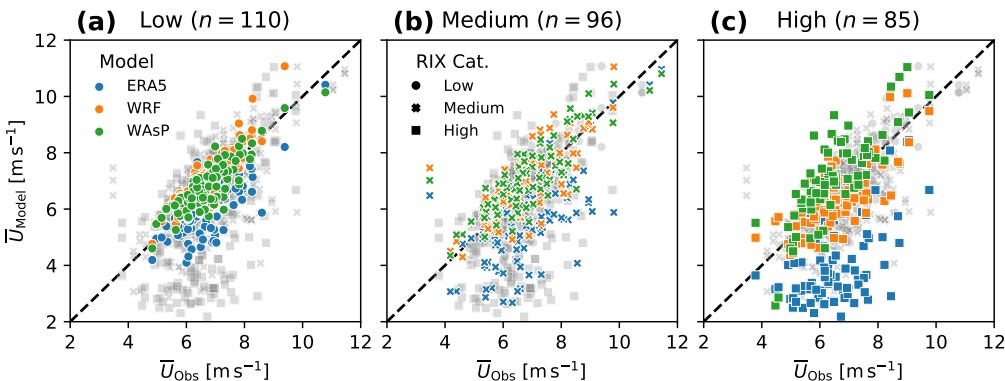

**Figure 8.** Observed versus modelled mean wind speed for ERA5, WRF, and WAsP. The three subplots include the same scatter points, but each of them highlights a different ruggedness index (RIX) category: low (a), medium (b), and high (c). The number of masts ($n$) in each category is indicated above the subplots.

and the WRF wind speeds become more negative with increased complexity. This is possibly due to under resolved orographic speed-up effects occurring at the more complex sites, since these masts tend to be placed on top of hills and ridges, where stronger wind is expected. For both ERA5 and WRF, the spread for biases in the mean wind speed are comparable between the samples in *medium* and *high* RIX classes, but slightly smaller for *high* RIX.

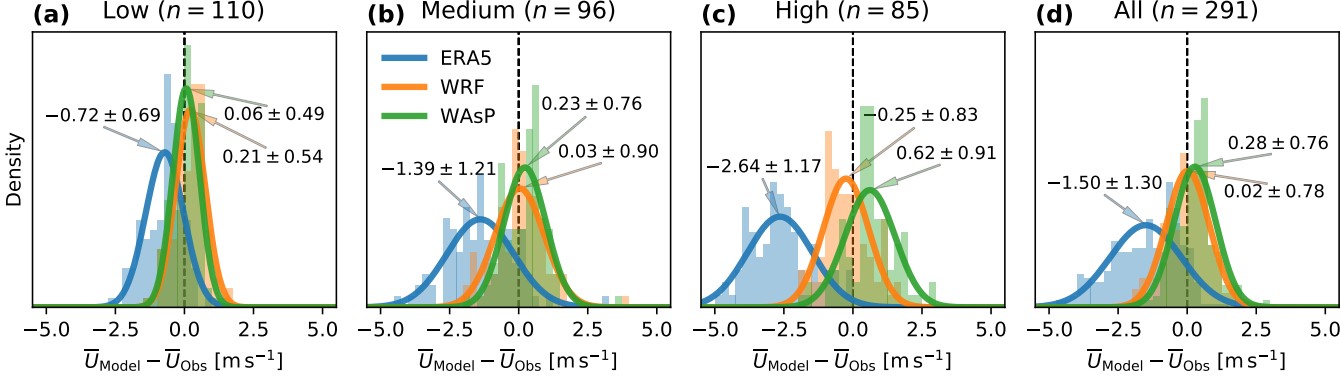

**Figure 9.** Distributions of wind speed biases ($\overline{U}_{\text{Model}} - \overline{U}_{\text{Obs}}$) for ERA5, WRF, and WAsP split by ruggedness index (RIX) category: low (a), medium (b), high (c), and all of the samples combined (d). Fitted normal distributions (lines) are annotated by the mean and standard deviation of the samples ($\mu \pm \sigma$). The number of masts ($n$) in each category is indicated above the subplots.

5    The mean biases in wind speed from WAsP and WRF are most similar in simple terrain, where the WAsP model makes the smallest adjustments to the WRF model wind climates (Fig. 9). The adjustments that are made by WAsP cause a reduction in the bias relative to the ones from WRF (from 0.21 to 0.06 m s$^{-1}$), and spread (from 0.54 to 0.49 m s$^{-1}$). The bias of the WAsP wind speeds (overestimation) and spread increase with increasing complexity, indicating that the linearised flow model



in WAsP has too large of an orographic wind speed speed-ups for many of the sites in medium and especially *high* RIX, as is expected.

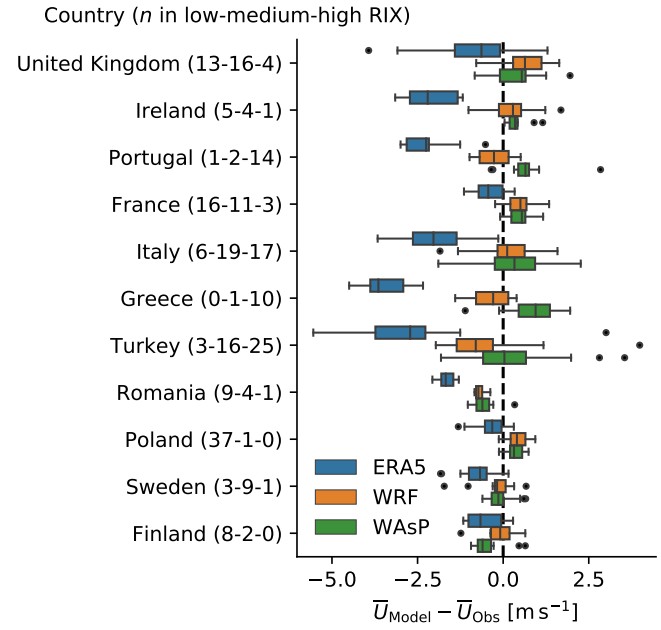

**Figure 10.** Boxplots of the distribution of mean wind speed biases ($\overline{U}_{\mathrm{Model}} - \overline{U}_{\mathrm{Obs}}$) by ERA5, WRF, and WAsP for the eleven countries containing most masts. The number of masts in each ruggedness index (RIX) category is shown in the parenthesis. The boxes indicate the $2^{\mathrm{nd}}$ and $3^{\mathrm{rd}}$ quartiles. Whiskers extents to $1.5\times$ the inter-quartile range (extent of $2^{\mathrm{nd}}$ and $3^{\mathrm{rd}}$ quartile) or to the outermost data point. Points indicate outliers outside the $1.5\times$ inter-quartile range.

The aggregate statistics of the mean wind speed biases presented in Fig. 9 do not show the spatial dependencies of the biases. But, some of these patterns are revealed in Fig. 10, which shows boxplots of the mean wind speed biases for the three stages of the model chain in the eleven countries with the most masts. The countries generally have low or high complexity, with few having an even mix. As expected, the figure shows that the bias and the spread of the biases are larger in countries with many sites in highly complex terrain, e.g. Turkey and Italy. However, significant differences in biases exist in some countries with mostly simple sites, e.g. the underestimation by WRF and WAsP in Romania and overestimation in Poland. These differences may be caused by biases associated with the large scale flow included in the ERA5 reanalysis data that WRF cannot correct, which influences the bias on a region scale as opposed to, for example, local influences from the terrain. Systematic biases in the measurements, e.g. from the same technical personnel and instrumentation at nearby clusters of masts, can not be ruled out either. The WAsP results generally follow the WRF results in simple terrain and deviate more in complex terrain, where the WAsP results tend to have larger wind speeds than those from WRF. Finland is a curious exception where the WAsP results have decreased the mean wind speed compared to those from WRF. The land use near the Finish sites is mostly dominated by water bodies, coniferous and mixed forests, and grasslands and pastures. Thus, the decrease may be related to an increase





in effective surface roughnesses in WAsP associated with a larger influence of forests or an overestimation of the surface roughness assigned to the forest classes.

Additional analysis (not shown) revealed that the mean wind speed biases of all three models are highly linked to spatial patterns. For ERA5 there is a tendency of reduced underestimation of the mean wind speed with latitude, with the largest

underestimations found in the south (Italy, Greece, Turkey in particular) and smaller, but still generally negative, biases found further to the north (Poland, France, Scandinavia in particular). In contrast, WRF shows a negative trend in mean wind speed error exists with longitude, with slightly larger negative biases found furthest to the east (Romania, Turkey, and Greece in particular) than to the west (UK, Ireland, and France in particular). The orographic complexity obviously has a strong spatial dependency as well, so latitude and longitude are not independent of RIX, which can explain some of this spatial dependency.

But other contributing factors also play a role, this could be, for example, spatially correlated biases related to large-scale patterns in the flow. For WAsP, RIX explains most of the variance of mean wind speed biases. This makes sense, bearing in mind the previous results, which shows how the wind speed speed-ups in orographically complex terrain cause large over-estimations.

WAsP mean wind speed biases tend to be larger for lower heights above the surface. Since this is not seen for WRF, it

suggests that the WAsP terrain or vertical extrapolation effects are overestimated closer to the surface. However, the tallest measurements are more frequent in low-RIX terrain, and smaller masts are more frequent in the high-RIX terrain, so some degree of collinearity exists between RIX and mast height, which could explain some of these differences.

Further analysis (not shown) also suggested that the variation of the surface roughness magnitude and complexity are not primary factors explaining the variation in mean wind speed bias for the three stages of the NEWA model-chain. Surface

roughness effects are, by definition, important for the magnitude of the wind speed at each site because of the momentum drain it exerts on the flow. Thus, the fact that it does not show any strong relation to the wind speed biases suggests that mean wind speed biases are not systematically associated with mischaracterisation of the effective surface roughness or of internal boundary layer effects.

### 3.2.2 Mean power generation biases

Ultimately, for the application of the NEWA wind climate estimates, the accuracy of the estimated power production is more important than capturing the mean wind speed. This requires the accurate simulation of the entire wind speed probability density function, and particularly the most critical wind speeds of the turbine-specific power curve, on the steepest part of the power curve.

To illustrate the accuracy of the NEWA wind climates for simulating the mean power generation of one specific turbine,

the NREL 5 MW power curve was used as an example (Fig. 11). The non-linearity of the power curve enhances the relative importance of a limited range of wind speeds of the wind speed PDF where the power curve is steepest (in this case 8–12 m s$^{-1}$). Since power curves can be quite different from turbine to turbine, the results for the NREL 5 MW turbine are not general, and could vary substantially depending on what turbine is used.





The mean power generation biases, in percentage, are defined as

$$\overline{P}_{\mathrm{Bias}} = 100 \times \frac{\overline{P}_{\mathrm{Model}} - \overline{P}_{\mathrm{Obs}}}{\overline{P}_{\mathrm{Obs}}}, \tag{1}$$

where $\overline{P}_{\mathrm{Model}}$ and $\overline{P}_{\mathrm{Obs}}$ in W are the estimated mean power generation from the model and the measurements, respectively, calculated using the full simulated wind speed distribution.

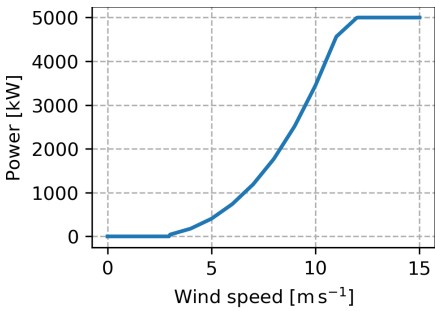

**Figure 11.** Power curve for the NREL 5 MW reference wind turbine (Jonkman et al., 2009).

The distributions of the biases in the mean wind power generation ($\overline{P}_{\mathrm{BIAS}}$) for the three stages of the model chain is shown in Fig. 12. The mean wind power generation is calculated as the sector-weighted mean from the binned wind distribution convoluted with the power curve. For the 291 masts, the underestimation of the wind speed in the ERA5 results in a large underestimation of the mean generated power, $-40.2 \pm 32.7\%$. This underestimation is especially large in *high* RIX terrain ($-69.4 \pm 24.9\%$). In *low* RIX terrain, the underestimation is also fairly large ($-20.5 \pm 20.5\%$). Since the percent biases are lower-bounded at -100 %, the ERA5 biases in the *high* RIX group do not follow the fitted Gaussian distribution well (the median value is $-78.14\%$). A Poisson distribution may describe the distribution better, but for simplicity the normal distribution is used regardless.

The power estimated using the WRF model winds has the lowest average error ($6.2 \pm 25.2\%$) across all masts. The distributions of power biases separated into groups by RIX class follow a similar pattern to the one seen for wind speed biases: the spread is smallest in simple (*low* RIX) orography ($11.3 \pm 18.3\%$), but the overall bias is larger than in *medium* ($7.4 \pm 29.1\%$) and *high* ($-1.8 \pm 26.2\%$) complexity sites.

For all the masts, the average mean power generation estimated from WAsP winds is overestimated by $13.3 \pm 27.4\%$. In *low* RIX terrain, a reduction of mean power bias and spread is seen compared to the WRF results, however the power is still overestimated ($7.3 \pm 17.6\%$ on average). In *medium* RIX terrain, just like for mean wind speed, the spread of mean power is reduced relative to WRF, but the average overestimation is enhanced by WAsP. In the most complex orography (*high* RIX), WAsP significantly overestimates the power and increases the spread of $\overline{P}_{\mathrm{Error}}$ ($21.2 \pm 36.7\%$) relative to the estimates made with WRF simulated winds.

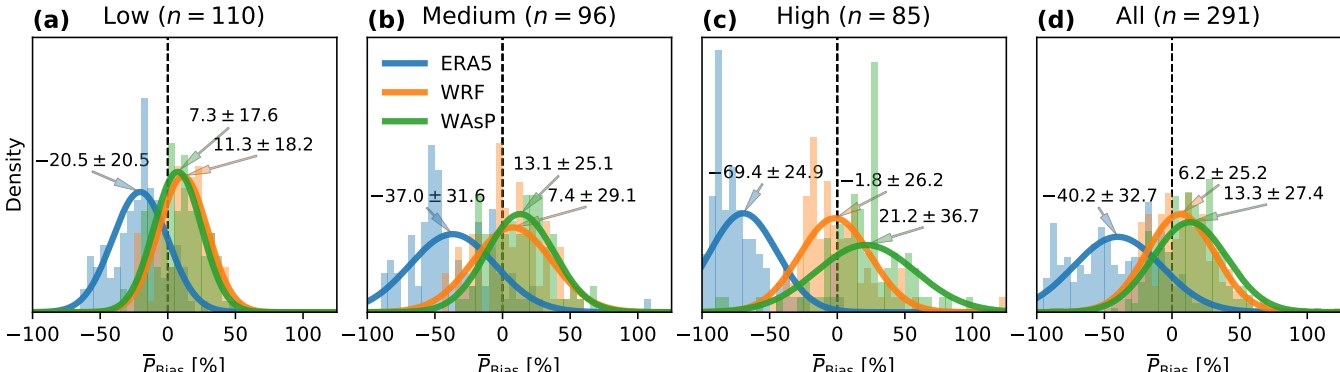

**Figure 12.** Distributions of the biases in mean generated power for the various stages of the model chain grouped by ruggedness index (RIX) category: low (a), medium (b), high (c), and all of the samples combined (d). The power was estimated by the modelled and observed wind climates combined with the NREL 5MW reference turbine power curve. Fitted normal distributions (lines) are annotated by the mean and standard deviation ($\mu \pm \sigma$). The number of masts ($n$) in each category is indicated above the subplots.

### 3.2.3 Wind direction biases

The earth movers distance (EMD) score between modelled and observed wind direction CDFs ($\text{EMD}_{\text{WD}}$) is used to evaluate the modelling biases in the wind direction distributions of the wind climates. This metric can generally be understood as the amount of physical work needed to move a pile of soil in the shape of one distribution to that of another distribution and is

5 introduced in more detail in part 1 of this study (see Section 4 in Hahmann et al., 2020b).

The $\text{EMD}_{\text{WD}}$ samples are not normally distributed, so a $\log(x)$ transformation was made. After the transformation the samples are close to normally distributed (Fig. 13).

The distributions of $\log(\text{EMD}_{\text{WD}})$ is shown in Fig. 13. These show that in *low* RIX terrain ERA5 estimates the wind direction distributions most accurately, on average, than the other stages of the model chain, but the spread for ERA5 is much greater

10 than for WRF and WAsP. The WRF and WAsP wind direction distributions produce very similar results in simple terrain, although a very slight improvement of WAsP is seen relative to WRF. In *medium* and *high* RIX terrain WRF ($8.85° \pm 2.00$ and $11.75° \pm 2.04°$respectively) estimates the wind direction distributions more accurately than ERA5 ($9.60° \pm 1.96°$and $16.01° \pm 1.98°$respectively). WAsP reduces the accuracy of WRF in *high* RIX terrain, possibly due to over corrections of the orographically induced turning of the wind in complex orography.

### 4 Discussion

The NEWA mesoscale model setup has advantages and disadvantages. The setup selection process was mainly driven by the model accuracy in terms of reproducing the wind speed distribution from a series of sensitivity experiments, but also by computational efficiency considerations (i.e. the domain size). These aspects are discussed in more detail in the Part 1 paper



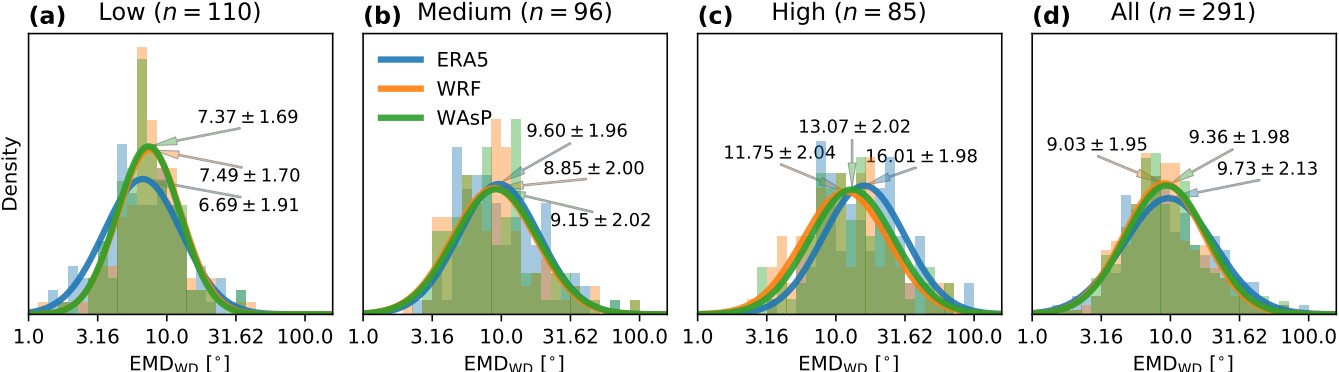

**Figure 13.** Distributions of the log-transformed samples of statistical distance ($\mathrm{EMD_{WD}}$) between the observed and modelled wind directions for the three stages of the model chain separated into groups by the ruggedness index (RIX) category: low (a), medium (b), high (c), and all the samples (d). The lines are fitted normal distributions that are annotated by the mean and standard deviation ($\mu \pm \sigma$) transformed back to physical units of degrees. The number of masts ($n$) in each category is indicated above each subplot.

of this study (Hahmann et al., 2020b). It is important to mention here that the data from the 291 mast used for evaluation were not used for the selection of the WRF model setup used in the NEWA production simulations.

The choice of the domain configuration is one of the most debatable aspects of the mesoscale setup. The choice of having each country (with exceptions, see Hahmann et al., 2020b) covered in one domain was in part motivated by the rationale that wind farm projects rarely cross borders and thus the points of interest for one wind farm should be in one domain only. This
should avoid the confusion for the end user that could result if inconsistent time series, which originate from two different domains, are used within one project. However, the "one-domain per country" criterion has limitations, for example some large-scale wind systems are generated or modified by orography that is contained in the neighbouring domain. This could be the reason for the lower wind speeds in WRF compared to ERA5 over the Aegean Sea (Fig. 7a) where the TR and GR domains
do not contain the orography of the other, which may have an impact on the Etesian wind system that dominates the flow in the region. More details and a comparison to satellite data can be found in Hasager et al. (2019).

The WAsP downscaling methodology is sensitive to the values of surface roughness length used, and their representativeness. In this study, the conversion from CORINE land-use classes to surface roughnesses relies on several key assumptions: (1) that the 44 classes accurately capture the real variability of land-use, and (2) that the mapping from land-use class to surface
roughness length is accurate. However, the accuracy of the maps and surface roughness is unknown. Previous studies have estimated the mean surface roughnesses uncertainties on the order of a factor of ~3 (Kelly and Jørgensen, 2017). Halving and doubling all the roughnesses in Table 2 leads to mean wind speed errors for the 291 masts in this study of $-0.11 \pm 0.75$ and $0.61 \pm 0.78$ after downscaling with WAsP, showing the wide span of results one can obtained within "reasonable" roughness values.
The traditional WAsP methodology, as opposed to that of WRF-WAsP method used here, involves predicting the wind climate at a target location from a nearby observed wind climate. Thus, the same terrain data is used to estimate correction




factors for the generalisation and the subsequent prediction at the target location. This means that any mischaracterisation of the terrain effects during the generalisation step could be partly compensated by corresponding and opposite errors during the prediction step. This compensation does not occur when the wind climate is derived from the WRF model simulations, which use and respond to surface roughness and terrain in a different way than the WAsP flow models. In NEWA, the parameters

used in the generalisation due to roughness changes (see Hahmann et al., 2019, for more details) were calibrated to generate smooth generalised wind climate estimates across flat coastal zones in northern Europe. No calibration was done for terrain speed up. The use of these parameters can introduce systematic biases in the generalisation of the WRF wind climatologies that are translated to the WAsP wind climatologies, but their nature is currently unknown.

Several wind atlases for regions or countries in Europe have been released in recent years (Tammelin et al., 2013; Weiter

et al., 2019; Kotroni et al., 2014; Wijnant et al., 2019). They can only be compared indirectly to the NEWA wind atlas because they represent different periods and have been evaluated against different observations (e.g. location, height a.g.l, and duration) than NEWA.

The Finnish wind atlas was validated against 20 met mast over a one year period with a mean positive bias of around $0.2$–$0.3\,\mathrm{m\,s^{-1}}$ in the mesoscale model (Tammelin et al., 2013). However, the masts only represent a fairly limited region of

the total atlas near Helsinki. The NEWA wind atlas was evaluated against 10 masts in Finland (distributed throughout the Country), and we showed (Fig. 10) that WRF generally had small mean wind speed biases over Finland, while ERA5 and WAsP mostly underestimate the mean wind speed slightly. Weiter et al. (2019) created a wind atlas of Germany by statistically correcting mesoscale wind fields for the impact of complex terrain on the flow. They found a relative bias of $10 - 25\,\%$ in power corresponding to $4\,\%$ to $10\,\%$ for the annual mean wind speed for most sites when evaluating the atlas against 12 wind

farms. The observations in our study only contained 4 masts within Germany. Thus, no comparison is possible between this study and the NEWA wind atlas.

The NEWA wind atlas was evaluated against 11 mostly mainland masts in Greece and generally showed near-zero bias or small underestimates of the mean wind speed for WRF (Fig. 10). A big improvement over ERA5, which underestimates the mean wind speed by $\approx 2.5$–$4\,\mathrm{m\,s^{-1}}$. Most of the masts (10 of 11) are in complex terrain and WAsP tends to overestimate the

wind speeds by $\approx 0$–$2\,\mathrm{m\,s^{-1}}$. Kotroni et al. (2014) made a numerical wind atlas for Greece based on the MM5 mesoscale model and evaluated it against six masts located across Greece. They found that the model overestimates the mean wind speed for mainland sites in Greece and underestimates it slightly for the island of Lemnos in the Aegean sea. However, the masts are all short (10 m or less), and thus are subject to large uncertainties due to the large variance near the surface and are not representative of the weather patterns at wind turbine hub heights.

Relatively small mean wind speed biases are seen for the NEWA WRF results in complex terrain, which is somewhat surprising given that the model does not resolve the complex orographic features. These small mean biases may be due to a compensation of errors: underrepresented speed-up effects on top of the hills and ridges, where the masts are placed, are partly compensated by a general overestimation of the wind speed over hills resulting from unresolved orographic drag (Jimenez and Dudhia, 2012). This could be investigated further by evaluating the results over a more evenly distributed network of masts,

which included valleys.


The NEWA wind atlas presents, for the first time, a comprehensive evaluation against a large dataset of tall masts, synchronised in time, and spanning most land regions covered by the atlas. There is, however, still significant limitations of the evaluation due to the relatively small number of available masts in the dataset. First, the database lacks masts in some regions that are key for wind energy, based on the number of installations and ongoing development, including Denmark, Germany,
Belgium, Spain, and the Netherlands. Second, masts are often clustered at prospect sites, which means that overall fewer regions are represented. Third, masts are positioned at wind energy relevant positions and not evenly distributed across different types of terrain, that is the top of hills and ridges are more frequently sampled than valleys and mountain basins. However, this is not all bad because the wind atlas is targeted for areas with possible wind energy development.

It is not feasible to evaluate the NEWA long-term wind atlas itself because the wind atlas represents the period between
1989 and 2018 and no measurements cover that entire period. In addition, measurement heights differ from mast to mast and rarely coincide with the fixed heights of the atlas. Therefore, the evaluation is based on mast-specific wind climates estimated with the NEWA model-chain, modified slightly to be flexible for the purpose of the evaluation. The main differences between the methods used for the long-term atlas and for the evaluation presented here are: (1) the evaluation method represents wind climates as a histogram (bins) throughout, while the long-term method fits a Weibull distribution during the generalisation
step; (2) the evaluation method assumes neutral stability for vertical extrapolation and makes no stability correction to the wind climate, while the long-term method assumes slightly stable conditions (on land) for vertical extrapolation and makes a stability correction (Kelly and Troen, 2016) during the generalisation step. These two factors should add very small differences between the two methods of evaluation.

In complex terrain, further downscaling of the WRF model wind climatologies via high-resolution dynamical flow models,
e.g. large-eddy simulation (LES) or unsteady Reynolds-Averaged Navier-Stokes (uRANS) models, is expected to improve the accuracy of the estimated wind climates (Sanz Rodrigo et al., 2017). However, this is at a much greater computational cost than the linearised flow model utilised for NEWA and currently cannot be done on a European scale even using modern super computers. However, for single complex terrain sites of interest these methods are already being applied (Duraisamy et al., 2014; Rodrigues et al., 2016; Olsen, 2018; Santoni et al., 2018; Barcons et al., 2019, e.g.). To improve the accuracy of future
wind atlases, it may thus be appropriate to separately downscale the wind climates in regions with higly complex terrain using a CFD model, while using more simple methods, like WAsP, for most regions.

## 5   Summary and conclusions

The NEWA wind atlas was created and released to the general public on 27 June 2019. The NEWA wind atlas provides a meso- and microscale wind climatology for the countries of the European Union plus Norway, Switzerland, the Balkans, and Turkey
that is based on simulations with the WRF mesoscale and WAsP microscale model. The mesoscale model simulations were forced by initial and boundary conditions from ERA5 and sea surface temperatures and sea-ice from OSTIA (Donlon et al., 2012). The atlas includes atmospheric and surface time series derived from the WRF model simulations for 30 years, with a half-hourly resolution at seven wind energy relevant heights and 3 km horizontal grid spacing. The microscale model layer



provides mean wind speed and power density at three levels (50, 100, and 150 m a.g.l.) at a horizontal grid spacing of 50 m. The simulations for the mesoscale database were conducted using a high degree of automation on the HPC cluster MareNostrum between April 2018 and March 2019 using a setup that was defined based on extensive sensitivity tests that are documented in the first part of this study (Hahmann et al., 2020b). The microscale downscaling was carried out on an HPC system at DTU.

The NEWA model-chain, which downscales the wind climatology from ERA5 to WRF to WAsP, was validated using wind measurements from 291 European tall masts. The model error was found to be related to the orographic complexity surrounding each mast, which was characterised using the ruggedness index (RIX). The main findings of the evaluation are:

- The average mean wind speed bias for the 291 masts is $-1.50 \pm 1.30 \, \mathrm{m \, s^{-1}}$ for ERA5, $0.02 \pm 0.78 \, \mathrm{m \, s^{-1}}$ for WRF, and $0.28 \pm 0.76 \, \mathrm{m \, s^{-1}}$ for WAsP.

- The results for the masts in simple orography shows that downscaling the WRF wind climates using WAsP, reduces the mean wind speed bias and spread: from $0.21 \pm 0.54 \, \mathrm{m \, s^{-1}}$ to $0.05 \pm 0.49 \, \mathrm{m \, s^{-1}}$ for WRF and WAsP, respectively

- For the masts located in the most complex orography, downscaling the WRF wind climates using WAsP resulted in large over estimations of the mean wind speed. This indicates that the microscale model, WAsP, is over estimating the orographic speed-up incurred by the steep terrain. This is a known behaviour of the WAsP linearised flow model, which
assumes attached flow and gentle slopes (Jackson and Hunt, 1975).

- The average mean power generation biases (using the NREL 5MW reference turbine power curve) for all masts are $-40.2 \pm 32.7\%$ (ERA5), $6.2 \pm 25.2\%$ (WRF), and $13.3 \pm 27.4\%$ (WAsP). The distribution of the biases among complexity classes is similar to that seen in the biases of the wind speed.

- The wind direction differences were quantified using the Earth Movers Distance (EMD). In simple orography ERA5
wind direction distributions are, on average, more accurate than WRF and WAsP. In complex orography the WRF wind direction distributions are most accurate, followed by WAsP, and then ERA5.

The New European Wind Atlas is publicly available for visual investigations and data download via: https://map.neweuropeanwindatlas. eu/. The wind atlas could be enhanced by additional layers for e.g. extreme winds and additional turbulence quantities. Further evaluation, especially in countries with limited of mast data in our study, and the computation of derived quantities could en-
hance the wind atlas product in the future. In general, more research is needed to improve the understanding of the relationships between modelled meso- and microscale winds and local measurements.

*Code availability.* The WRF model code is open source and can be obtained from the WRF Model User's Page (http://www2.mmm.ucar.edu/wrf/users/, doi:10.5065/D6MK6B4K). It should be installed following the general instructions given there. In the NEWA production run, we used WRF version 3.8.1 and modified it as described in Hahmann et al. (2020b). The code modifications as well as namelists, tables and
domain files we used are available from the NEWA GitHub repository: https://github.com/newa-wind/Mesoscale and permanently indexed in Zenodo (Hahmann et al., 2020a).



*Data availability.* The resulting NEWA data is available from https://map.neweuropeanwindatlas.eu/

The surface and forcing data used in the mesoscale simulations are publicly available:

ERA5: https://climate.copernicus.eu/climate-reanalysis,

OSTIA: http://marine.copernicus.eu/services-portfolio/access-to-products/?option=com_csw&view=details&product_id=SST_GLO_SST_

L4_NRT_OBSERVATIONS_010_001,

CORINE: https://land.copernicus.eu/pan-european/corine-land-cover,

ESA-CCI: http://cci.esa.int/data.





## Appendix A: Mesoscale wind atlas parameters

**Table A1.** Overview of the final wind atlas parameters that were stored in netCDF files following CF-1.6 conventions. The four-dimensional variables are given at 50, 75, 100, 150, 200, 250 and 500 m a.g.l. All three- and four-dimensional variables are provided at 30-min time intervals. A post-processing script (written in python) that calculates these parameters is given in: https://github.com/newa-wind/Mesoscale

| shortname | longname | units | dimensions |
|---|---|---|---|
| T | Air temperature | K | 4 |
| TKE | Turbulent kinetic energy | $\mathrm{m^2\,s^{-2}}$ | 4 |
| WS | Wind speed | $\mathrm{m\,s^{-1}}$ | 4 |
| WD | Wind direction | ° | 4 |
| PD | Power density | $\mathrm{W\,m^{-2}}$ | 4 |
| QVAPOR | Water vapour mixing ratio | 1 | 4 |
| ABLAT_CYL | Ice ablation on standard cylinder | kg | 3 |
| ACCRE_CYL | Ice accretion on standard cylinder | kg | 3 |
| HFX | Surface sensible heat flux | $\mathrm{W\,m^{-2}}$ | 3 |
| LH | Surface latent Heat Flux | $\mathrm{W\,m^{-2}}$ | 3 |
| PRECIP | Precipitation rate | $\mathrm{kg\,m^2}$ | 3 |
| PBLH | PBL height | m | 3 |
| PSFC | Surface pressure | Pa | 3 |
| Q2 | Specific humidity at 2m | 1 | 3 |
| RHO | Air density | $\mathrm{kg\,m^{-3}}$ | 3 |
| RMOL | Inverse Obukhov length | $\mathrm{m^{-1}}$ | 3 |
| SEAICE | Sea ice fraction | 1 | 3 |
| SWDDNI | Shortwave direct normal radiation | $\mathrm{W\,m^{-2}}$ | 3 |
| SWDDIR | Shortwave diffuse incident radiation | $\mathrm{W\,m^{-2}}$ | 3 |
| T2 | Air temperature at 2 m | K | 3 |
| TSK | Surface skin temperature | K | 3 |
| UST | Friction velocity | $\mathrm{m\,s^{-1}}$ | 3 |
| WD10 | Wind direction at 10 m | ° | 3 |
| WS10 | Wind speed at 10 m | $\mathrm{m\,s^{-1}}$ | 3 |
| ZNT | Surface aerodynamic roughness length | m | 3 |
| ALPHA | map projection distortion | ° | 2 |
| HGT | Surface elevation | m | 2 |
| LANDMASK | Landmask (1 for land, 0 for water) | 1 | 2 |
| LU_INDEX | Dominant land use category (USGS) | - | 2 |
| XLAT | Center latitude of grid cell | ° | 2 |
| XLON | Center longitude of grid cell | ° | 2 |
| Times | Time | UTC | 1 |





## Appendix B: Land use and roughness length conversion

**Table B1.** Look-up table for the surface roughness length as a function of the USGS land use category in the NEWA and default NCAR WRF model configuration. Only values changed from default are shown.

| USGS type | land-use land cover class | $z_0$ NEWA [m] | $z_0$ WRF orig range [m] |
|---:|---|:---:|:---:|
| 2 | Dryland Cropland and Pasture | 0.10 | 0.05–0.15 |
| 3 | Irrigated Cropland and Pasture | 0.10 | 0.02–0.10 |
| 4 | Mixed Dryland/Irrigated Cropland and Pasture | 0.10 | 0.05–0.15 |
| 5 | Cropland/Grassland Mosaic | 0.10 | 0.05–0.14 |
| 7 | Grassland | 0.10 | 0.10–0.12 |
| 8 | Shrubland | 0.12 | 0.01–0.05 |
| 9 | Mixed Shrubland/Grassland | 0.12 | 0.01–0.06 |
| 11 | Deciduous Broadleaf Forest | 0.90 | 0.5 |
| 12 | Deciduous Needleleaf Forest | 0.90 | 0.5 |
| 13 | Evergreen Broadleaf Forest | 0.90 | 0.5 |
| 14 | Evergreen Needleleaf Forest | 0.90 | 0.5 |
| 15 | Mixed Forest | 0.50 | 0.20–0.50 |
| 17 | Tidal zone[a] | 0.001 | 0.20 |

[a] Originally called "Herbaceous Wetland" in the default WRF vegetation table.

*Author contributions.* MD and BTO wrote the first draft and took over responsibility in automatising the mesoscale (MD) and microscale simulations as well as the evaluation with tall masts (BTO). MD, BTO, BW, ANH, JoB, YE, EGB, JFGR, JN, MSM and TS conducted simulations for the NEWA mesoscale wind atlas, NND and WT automatised and optimised the microscale and mesoscale simulations on the HPC systems. MZ participated in the tall mast evaluation and discussions. JB contributed on aspects of the model-chain downscaling to microscale, interpretation of the evaluation and defining atlas outputs. All authors participated in the writing, editing and internal reviews of the manuscript.

*Competing interests.* The WAsP software that was used to produce the NEWA microscale wind atlas is developed and commercially sold by DTU Wind Energy.



*Acknowledgements.* The European Commission (EC) partly funded NEWA (*NEWA- New European Wind Atlas*) through FP7 (topic FP7-ENERGY.2013.10.1.2). The authors of this paper acknowledge the support from the Federal Ministry for the Economic Affairs and Energy, on the basis of the decision by the German Bundestag (Germany - ref. no. 0325832A/B); the Danish Energy Authority (EUDP 14-II, 64014-0590, Denmark); Latvijas Zinatnu Akademija (Latvia); the Ministerio de Economía y Competitividad (Spain -refs. no. PCIN-2014-

017-C07-03, PCIN-2016-176, PCIN-2014-017-C07-04, PCIN-2016-009, PCIN-2014-013-C07-04 and PCIN-2016-080); the Scientific and Technological Research Council of Turkey (Turkey-grant number 215M386). MSM additionally acknowledges the Spanish Ministerio de Educación, Cultura y Deporte support through "José Castillejo" Fellowship (reference CAS18/00316). The computing resources for the calculation of the mesoscale wind atlas were made available through a PRACE grant (project number 2017174128) between April 2018 and March 2019. The microscale atlas was computed between April 2019 and June 2019 on the DTU "Sophia" HPC cluster. A feasibility study

and many tests for the production run were carried out on the HPC cluster EDDY located at the University of Oldenburg and funded by the Federal Ministry for Economic Affairs and Energy under grant number 0324005. Access to the tall mast data used for the evaluation has kindly been granted by Vestas Wind Systems A/S. The Authors are grateful to Yavor Hristov from Vestas for helping in making the mast data available. The WRF model simulations were initialised using ERA5 and OSTIA data downloaded from ECMWF and Copernicus Climate Change Service Climate Data Store and Copernicus Marine Environment Monitoring Service. The authors are grateful to Mark Kelly from

DTU Wind energy for valuable comments and discussions, and to the technical staff at PRACE, DTU and the University of Oldenburg for assistance and maintenance of the systems, and to Nazka Maps for making the NEWA wind atlas website. Data processing and visualisation for this study was made using the python programming language and involved extensive use of the following software packages: numpy (Oliphant, 2006; Van Der Walt et al., 2011), scipy (Jones et al., 2001–), pandas (McKinney, 2010), dask (Dask Development Team, 2016), xarray (Hoyer and Hamman, 2017), matplotlib (Hunter, 2007), scikit-learn (Pedregosa et al., 2011). The authors are grateful for the tools

provided by the open source community, which has benefited the making of this study tremendously.



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
