# Peer review of "The Making of the New European Wind Atlas, Part 2: Production and Evaluation"

_Geoscientific Model Development, 2020_

## Referee Comment (RC1) · Anonymous Referee #1 · 22 May 2020

This manuscript presents the results of a major multi-institutional technical landmark project to develop a new European wind energy atlas. The work is of the highest caliber, and for the most part is completely and carefully described. The evaluation of the wind atlas results demonstrates that the atlas provides usefully accurate estimates of the annual wind energy resource in both areas of flat and complex terrain. Notably the manuscript highlights the assumptions, simplifications, and limitations that were necessary to construct the atlas. The paper, when published, will be extremely useful to all who make use of the new atlas.

Specific comments:

P8. 8, Line 26: "The effective horizontal resolution of the mesoscale atlas is several kilometers . . ." Is this referring to the effective resolution of the 3 km WRF runs, or of

the final atlas after downscaling? If it is of the 3 km WRF simulations, then I would say "resolution of the finest grid WRF simulations is". Also the effective resolution according to Skamarock (2004) will be approximately 7$\Delta$x, or ∼20 km, which is more than "several km".

Pg. 9, Lines 4-12. The procedure used here seems to be that WRF is run to simulate flow over topography, the effects of the topography then are removed using a complex procedure including the use of a simplified linear model, and then the effects of the topography are added back in using another linear model. The authors should motivate and describe in general terms why such a convoluted process is necessary.

Also the link to the reference by Hahmann et al., 2014 that describes this methodology already gives "page not found".

Pg. 10. Do the surface roughness lengths vary depending on season? Won't the roughness lengths for cropland and deciduous forests be much different from summer to winter? If they are not seasonally varying, are they more representative of summer or winter?

Pg. 13, line 32. How were the data on opposing booms used to reduce flow distortion effects? Presumably some sort of criteria were used to select one boom over the other. What were these criteria? Were the criteria dependent on wind direction, or only on speed differences? Was the selection based on hourly data, 1-min data, etc?

Pg. 15, lines 4-14. "For evaluation of the downscaled wind climate at each mast site, some modifications to the WRF-WAsP methodology were made" and following sentences. If I read this correctly, a separate and different set of downscaling procedures has been applied only to those locations where the atlas is compared to the observations. Why do something different at the evaluation sites? Won't this be comparing winds that are different from the rest of the atlas? If there are advantages to this new methodology, why not apply it everywhere?

Technical corrections:

Pg. 2, Line 9: "It documents the meteorological basis for large parts of Europe". The meteorological basis of what? Of the atlas?

Pg.3, lines 2-8. If I interpret this correctly, the NEWA actually consists of two separate wind atlases. One is a mesoscale atlas based on WRF, while the second is a downscaled WRF-WaSP product. It this interpretation is correct, I suggest that the text be modified to clearly state this.

Pg. 4, Line 9: "while the nudging" should be "while nudging"

Pg. 18, line 7. Does "mean wind speed" refer to the annual mean?

―――――――――――――――

---

## Referee Comment (RC2) · Anonymous Referee #2 · 27 May 2020

This paper is of great interest to the community as it describes the details of how the New European Wind Atlas was composed, including the specifics of the modeling, and verifies the model results with data. The methods described are reasonable and state-of-the-science. Although WASP is not perfect, it's imperfections are well documented and fairly analyzed and discussed here.

Technical comments: - Detailed discussion of surface roughness and the difficulties is interesting. Thanks for including. - p. 14 , line 6 mentions that results were checked for obvious errors "like icing". On p. 4 line 5, you mention that additional code added to WRF estimates ice accumulation. This appears to be inconsistent. - p. 18 - Assume neutral atmospheric stability. This could be a large, inappropriate assumption. On line 15, authors mention that this assumption may cause small discrepancies. I would

be much more concerned since stable conditions, which in some locales occur nearly nightly, can cause low level jets, which can result in shear across the turbine blade as well as large errors in the wind speed. I guess it all depends on how well WRF models those. This could be discussed a bit more. - Very nice discussion of bias and consideration of RIX implications - The authors assume the NREL 5MW turbine for power estimates. Was there any opportunity to compare to actual power for a few existing farms? That would certainly provide a bit more confidence in power estimates. - p. 23, line 13 - interesting that WASP reduces the accuracy of WRF in high RIX terrain. Have you considered replacing the WASP results with WRF in those locales? Would be interesting to discuss. - p. 24, lines 3-11 - very nice analysis. p. 26, second paragraph - nice discussion of limitations. This is very helpful. - p. 27 - nice list of bullets. The final one discussed which models are more accurate in different orography. I'm confused then which model is shown on the website fpr wind direction. Is it always WASP? Or is it the most accurate model (WRF for complex, ERA5 for simple)? Which should be shown? Similar questions for bullet 3 for wind speed.

Minor comments: p. 6, lines 10-11 - "however" used twice in one sentence p. 8, line 8 - data WERE is appropriate. Please use "data" as plural throughout. There is mixed use in this manuscript - please change to be consistent. p. 9, line 14 - not a sentence p. 13 - line 18 - would likely have complex structures "in the flow" due to ... Please specify to help readers p. 14, lines 11 and 12 - data "were" - correct 3 times please. Several others later so won't point out each one. p. 21, line 18 - results, which "show" ... (agreement) p. 22, line 7 - Do you mean "convolved" rather than "convoluted"? p. 25, lines 23-24 - "A big improvement ....." not a sentence. p. 30, MD and BTO - likely "automating". automatising is not common English usage.

Please also note the supplement to this comment:
https://www.geosci-model-dev-discuss.net/gmd-2020-23/gmd-2020-23-RC2-supplement.pdf
* * *

---

## Author Comment (AC1) · 17 Jul 2020

The reviewers' comments are in black and our responses in blue.

**1    Response to Referee #2**

This paper is of great interest to the community as it describes the details of how the New European Wind Atlas was composed, including the specifics of the modeling, and verifies the model results with data. The methods described are reasonable and stateof- the-science.  Although WASP is not perfect, it's imperfections are well documented and fairly analyzed and discussed here.

Thank you for your thorough evaluation and very positive feedback to our manuscript. We answer all of your comments separately below.

**1.1 Technical Comments**

1. Detailed discussion of surface roughness and the difficulties is interesting. Thanks for including.

   Thank you.

2. p. 14, line 6. mentions that results were checked for obvious errors "like icing". On p. 4 line 5, you mention that additional code added to WRF estimates ice accumulation. This appears to be inconsistent.

   The manuscript needs clarification here. This refers to two separate things. An icing parameterization, i.e. risk of icing on turbine blades, was added to the WRF model code. The line above refers to the quality control of measurements and potential influence of icing. We propose to clarify this in the manuscript.

3. p. 18 - Assume neutral atmospheric stability. This could be a large, inappropriate assumption. On line 15, authors mention that this assumption may cause small discrepancies. I would be much more concerned since stable conditions, which in some locales occur nearly nightly, can cause low level jets, which can result in shear across the turbine blade as well as large errors in the wind speed. I guess it all depends on how well WRF models those. This could be discussed a bit more.

   It is important to stress that the assumption of neutral stratification only applies to the vertical extrapolation of the mean profile from the WRF output to the height of the instrument, which, in most cases, is a distance less than 12.5 m (up or down) and is taking place at elevations above 40 m.

Since WAsP is a statistical model, it cannot capture Low-level Jet's (LLJs). There-
fore, effects of LLJs are only present in the wind climates insofar as WRF has
captured them correctly.

4. Very nice discussion of bias and consideration of RIX implications

Thank you.

5. The authors assume the NREL 5MW turbine for power estimates. Was there
any opportunity to compare to actual power for a few existing farms? That would
certainly provide a bit more confidence in power estimates.

We agree that this would be valuable. However, the scope of the paper is al-
ready quite extensive, so we decided to focus on comparison with wind speed
measurements from masts, which avoids additional complexities associated with
comparison against power production. We suggest to put this as a suggestion for
possible future work at the end of the manuscript.

6. p. 23, line 13. - interesting that WASP reduces the accuracy of WRF in high
RIX terrain. Have you considered replacing the WASP results with WRF in those
locales? Would be interesting to discuss

Good point. We provide access to both the WRF-based and WAsP-based atlases
and a map of the RIX values on the website. Users can use the atlas best suited
for the locales based on the RIX map. We agree that it would be great to make
this filtering process easier for users in the future.

7. p. 24, lines 3-11. - very nice analysis, p. 26, second paragraph - nice discussion
of limitations. This is very helpful.

Thanks

8. p. 27 - nice list of bullets. The final one discussed which models are more
accurate in different orography. I'm confused then which model is shown on the

website for wind direction. Is it always WASP? Or is it the most accurate model (WRF for complex, ERA5 for simple)? Which should be shown? Similar questions for bullet 3 for wind speed.

For now, wind direction statistics are not displayed on the website and only the WRF wind speed time series are downloadable. We would like to also make the the WAsP based data more accessible, so these results should be seen as a documentation of the model results for future reference.

**1.2 Minor comments**

1. p. 6, lines 10-11 - "however" used twice in one sentence

   Thanks you. Corrected.

2. p. 8, line 8 - data WERE is appropriate. Please use "data" as plural throughout. There is mixed use in this manuscript - please change to be consistent.

   Thanks you. Corrected.

3. p. 9, line 14 - not a sentence

   Thanks you. Corrected.

4. p. 13 - line 18 - would likely have complex structures "in the flow" due to ... Please specify to help readers

   Thanks you. Corrected.

5. p. 14, lines 11 and 12 - data "were" - correct 3 times please. Several others later so won't point out each one.

   Thanks you. Corrected.

6. p. 21, line 18 - results, which "show" ... (agreement)

Thanks you. Corrected.

7. p. 22, line 7 - Do you mean "convolved" rather than "convoluted"?

Yes. Thank you. Corrected.

8. p. 25, lines 23-24 - "A big improvement ....." not a sentence.

Thank you. Corrected.

9. p. 30, MD and BTO - likely "automating". automatising is not common English usage.

Thank you. Corrected.

**1.3 Additional Clarifications**

We have received further non-documented feedback and spotted ourselves the following issues that we would like to clarify in the revised manuscript:

- Pg. 10 - line 22. Table B1 does not exist in Hahmann et al (2020) [1], it is B1 in the appendix of the second (this!) manuscript

- Pg. 24 - Figure 13. is not based on circular statistics in the submitted version of the manuscript, we have recomputed the metric and redrawn the figure without need for re-interpretation of the results

- We have added a discussion about optimising the wind atlas for the wind climate (distributions) instead of the accuracy of the time series.

References

[1] Andrea N. Hahmann et al. "The Making of the New European Wind Atlas, Part1: Model Sensitivity". In: Geosci. Model Dev. Discuss.2020 (2020).doi:10.5194/gmd-2019-349

---

## Author Comment (AC2) · 17 Jul 2020

**1   Response to Referee #1**

This manuscript presents the results of a major multi-institutional technical landmark project to develop a new European wind energy atlas. The work is of the highest caliber, and for the most part is completely and carefully described. The evaluation of the wind atlas results demonstrates that the atlas provides usefully accurate estimates of the annual wind energy resource in both areas of flat and complex terrain. Notably the manuscript highlights the assumptions, simplifications, and limitations that were necessary to construct the atlas. The paper, when published, will be extremely useful to all who make use of the new atlas.

[Figure]

Thank you for your positive feedback to our manuscript. We respond to all of your comments separately below.

**1.1 Specific Comments**

1. Pg. 8, line 26. "The effective horizontal resolution of the mesoscale atlas is several kilometers . . ." Is this referring to the effective resolution of the 3 km WRF runs, or of the final atlas after downscaling? If it is of the 3 km WRF simulations, then I would say "resolution of the finest grid WRF simulations is". Also the effective resolution according to Skamarock (2004) will be approximately $7\,\Delta x$, or $\approx$ 20 km, which is more than "several km".

   Good point. We suggest to avoid the term "resolution", and instead write: "The horizontal grid-spacing of the mesoscale atlas is three kilometres (in each direction)".

2. Pg. 9, lines 4–12. The procedure used here seems to be that WRF is run to simulate flow over topography, the effects of the topography then are removed using a complex procedure including the use of a simplified linear model, and then the effects of the topography are added back in using another linear model. The authors should motivate and describe in general terms why such a convoluted process is necessary. Also the link to the reference by Hahmann et al., 2014 that describes this methodology already gives "page not found".

   Thanks for spotting the wrong reference, which should have been Badger et al. 2014 [1]

   We agree that more motivation could be provided for readers outside the wind energy community. Due to the realtively coarse resolution of the WRF model topography, a downscaling with a microscale model is needed to provide more accurate wind data in particular in non-homogeneous terrain. For this purpose,

as indicated in the introduction, linearized flow models, such as the WAsP model, are well known and have extensively been used within the wind energy sector for site assessment in the past 30 years. The underlying principle is to estimate the wind climate at a point by vertically extrapolating from a known nearby wind climate, either measured or modelled. The extrapolation is done by first "removing" topographic effects from the known wind climate, estimated by the linearized flow model from the best available topographical maps. Then, vertical extrapolation is done using drag-law relations. Finally, the topographic effects (estimated by the same model) at the target point are added to the wind climate. This concept only works for short distances where the geostrophic forcing is similar.

The WAsP linearized flow model has also been used successfully in combination with WRF for several regional and global wind atlases, such as the Global Wind Atlas (GWA) and the Wind Atlas for South Africa (WASA) [2, 3, 6]. The principle of e.g. WRF-WAsP downscaling is the same as above, but instead of removing and adding topographical effects estimated from the same maps, the WRF elevation and land-use is used for "removing" topographical effects and the best available maps are used for "adding" topographical effects.

3. Pg. 10. Do the surface roughness lengths vary depending on season? Won't the roughness lengths for cropland and deciduous forests be much different from summer to winter? If they are not seasonally varying, are they more representative of summer or winter?

The minimum and maximum value of the surface roughness length in the vegetation table are the same; thus an annual cycle on the vegetation is not active. The constant value represent in most cases the geometric average. This is described in the companion paper [4]. Using a constant value of surface roughness facilitates the process of microscale downscaling.

4. Pg. 13, line 32. How were the data on opposing booms used to reduce flow

distortion effects? Presumably some sort of criteria were used to select one boom over the other. What were these criteria? Were the criteria dependent on wind direction, or only on speed differences? Was the selection based on hourly data, 1-min data, etc?

In general, no correction was made to reduce flow distortion in the measurements beyond existing corrections from the data provider. Only in few instances where two cups or sonic anemometers were present at the same height, but at different boom angles, was selective sampling used to combine the measurements and minimise distortion. We expect flow distortion to have some influence on the results, which, based on the impact we saw of including the correction in [4] and indications by e.g. [7], could be up to a few percent difference in annual mean wind speed.

Filtering was done on 10min averages.

5. Pg. 15, lines 4–14. "For evaluation of the downscaled wind climate at each mast site, some modifications to the WRF-WAsP methodology were made" and following sentences. If I read this correctly, a separate and different set of downscaling procedures has been applied only to those locations where the atlas is compared to the observations. Why do something different at the evaluation sites? Won't this be comparing winds that are different from the rest of the atlas? If there are advantages to this new methodology, why not apply it everywhere?

From the onset of the NEWA atlas production, we choose to use the default WRF-WAsP methodology for long-term mean wind climates making up the atlas. This method is well documented, computationally performant, and in most cases the assumed Weibull distributions should capture the 30-year wind climates well.

Our rationale for not assuming Weibull distributions for the validation, was to avoid parameterization biases that is expected to be exaggerated for these shorter one-year periods, compared to the long-term mean wind climate of final atlas. While

the histogram-based approach means we cannot use the stability correction from [5] (since it works on Weibull-parameters), we assumed that this difference would be smaller than the parameterization biases caused by assuming Weibull distributions.

We agree that we should give a better indication of the quantitative difference between using the two approaches and propose to add this information to the revised manuscript.

**1.2 Technical Corrections**

1. Pg. 2, line 9: "It documents the meteorological basis for large parts of Europe". The meteorological basis of what? Of the atlas?

   The meteorological basis for locations across most of Europe. "Meteorological basis" should be understood as the mean wind climate.

2. Pg. 3, lines 2–8. If I interpret this correctly, the NEWA actually consists of two separate wind atlases. One is a mesoscale atlas based on WRF, while the second is a downscaled WRF-WaSP product. It this interpretation is correct, I suggest that the text be modified to clearly state this.

   We agree that it can be made more explicit in the text that the NEWA Wind Atlas consists of two individual "atlases" or "layers" (mesoscale and microscale).

3. Pg. 4, line 9. "while the nudging" should be "while nudging"

   Thank you, corrected.

4. Pg. 18, line 7. Does "mean wind speed" refer to the annual mean?

   Yes, it refers to the mean of the validation period, which is up to a year.

**1.3  Additional Clarifications**

We have received further non-documented feedback and spotted ourselves the following issues that we would like to clarify in the revised manuscript:

- Pg. 10 - line 22. Table B1 does not exist in Hahmann et al (2020) [4], it is B1 in the appendix of the second (this!) manuscript

- Pg. 24 - Figure 13. is not based on circular statistics in the submitted version of the manuscript, we have recomputed the metric and redrawn the figure without need for re-interpretation of the results

- We have added a discussion about optimising the wind atlas for the wind climate (distributions) instead of the accuracy of the time series.

**1.4  References**

[1] Jake Badger et al.: Wind-climate estimation based on mesoscale and microscale modeling: Statistical-dynamical downscaling for wind energy applications. In: J. Appl. Meteorol. Clim. 53.8 (Aug. 2014), pp. 1901–1919. issn: 1558-8424. doi: http://dx.doi.org/10.1175/JAMC-D-13-0147.1.

[2] Andrea N. Hahmann et al.: Mesoscale modeling for the wind atlas for South Africa (WASA) Project. Tech. rep. TR-0050. last accessed: 19.10.2019. DTU Wind Energy, 2014, p. 77. http://orbit.dtu.dk/services/downloadRegister/107110172/DTU_Wind_Energy_E_0050.pdf.

[3] Andrea N. Hahmann et al.: Mesoscale Modelling for the Wind Atlas of South Africa (WASA) Project Phase II. English. Tech. rep. E-0188. Denmark: DTU Wind Energy, 2018. https://orbit.dtu.dk/files/192964222/DTU_Wind_Energy_E_0188.pdf.

[4] Andrea N. Hahmann et al.: The Making of the New European Wind Atlas, Part 1: Model Sensitivity". In: Geosci. Model Dev. Discuss. 2020 (2020). doi: http://dx.doi.org/10.5194/gmd-2019-349.

[5] M. Kelly and I. Troen.: Probabilistic stability and 'tall' wind profiles: theory and method for use in wind resource assessment. In: Wind Energy 19 (2016), pp. 227–241. doi: http://dx.doi.org/10.1002/we1829.

[6] Niels G Mortensen, Jens Carsten Hansen, and Mark C. Kelly. Wind Atlas for South Africa (WASA) Western Cape and parts of Northern and Eastern Cape Observational Wind Atlas for 10 Met. Masts in Northern, Western and Eastern Cape Provinces. Tech. rep. April. last accessed: 19.10.2019. DTU Wind Energy, 2014. https://orbit.dtu.dk/ws/files/110948908/DTU_Wind_Energy_E_0072.pdf.

[7] A. Westerhellweg, T. Neumann, and V. Riedel. FINO1 Mast Correction". 2012. https://pdfs.semanticscholar.org/cf85/2b7bc731b071162e537edf45f9578f4ec86e.pdf.